



# Trends, composition, and sources of carbonaceous aerosol in the last 18 years at the Birkenes Observatory, Northern Europe

Karl Espen Yttri[1], Francesco Canonaco[2,7], Sabine Eckhardt[1], Nikolaos Evangeliou[1], Markus Fiebig[1], Hans Gundersen[1], Anne-Gunn Hjellbrekke[1], Cathrine Lund Myhre[1], Stephen Matthew Platt[1], André S. H. Prévôt[2], David Simpson[3,4], Sverre Solberg[1], Jason Surratt[5], Kjetil Tørseth[1], Hilde Uggerud[1], Marit Vadset[1], Xin Wan[6], and Wenche Aas[1]

[1] NILU - Norwegian Institute for Air Research, P.O. Box 100, N-2027 Kjeller, Norway

[2] Paul Scherrer Institute (PSI), 5232 Villigen-PSI, Switzerland

[3] EMEP MSC-W, Norwegian Meteorological Institute, Oslo, Norway

[4] Department of Earth & Space Sciences, Chalmers Univ. Technology, Gothenburg, Sweden

[5] Department of Environmental Sciences & Engineering, University of North Carolina

[6] Key Laboratory of Tibetan Environment Changes and Land Surface Processes, Institute of Tibetan Plateau Research, Chinese Academy of Sciences Courtyard 16, Lin Cui Road, Chaoyang District, Beijing 100101, P.R. China

[7] Datalystica Ltd., 5234 Villigen, Switzerland

*To whom correspondence should be addressed:* Karl Espen Yttri, e-mail address: key@nilu.no

## Abstract

We present 18 years (2001–2018) of aerosol measurements: organic- and elemental carbon (OC and EC), organic tracers (levoglucosan, arabitol, mannitol, trehalose, glucose, 2-methyltetrols), trace elements and ions -at the Birkenes Observatory (Southern Norway), a site representative of the Northern European region. The OC/EC (2001–2018) and the levoglucosan (2008–2018) time series are the longest in Europe, with OC/EC available for the $PM_{10}$, $PM_{2.5}$ (fine) and $PM_{10-2.5}$ (coarse) size fractions, providing the opportunity for a nearly two-decade long assessment. Using positive matrix factorisation (PMF) we identify six carbonaceous aerosol sources at Birkenes: Mineral dust dominated (MIN), traffic/industry-like (TRA/IND), short range transported biogenic secondary organic aerosol ($BSOA_{SRT}$), primary biological aerosol particles (PBAP), biomass burning (BB), and ammonium nitrate dominated ($NH_4NO_3$), and one low carbon fraction, sea salt (SS).

We observed significant ($p<0.05$), large decreases of EC in $PM_{10}$ (-3.9% $yr^{-1}$) and $PM_{2.5}$ (-4.2% $yr^{-1}$), and a smaller decline in levoglucosan (-2.8% $yr^{-1}$), suggesting that OC/EC from traffic and industry is decreasing, while abatement of OC/EC from biomass burning has been slightly less successful. EC abatement of anthropogenic sources is further supported by decreasing EC fractions in $PM_{2.5}$ (-4.0% $yr^{-1}$) and $PM_{10}$ (-4.7% $yr^{-1}$). PMF apportioned 72% of EC to fossil fuel sources, further supported by PMF applied to absorption photometer data, which yielded a two-factor solution with a low aerosol Ångstrøm exponent (AAE=0.93) fraction assumed to be equivalent black carbon from fossil fuel combustion ($eBC_{ff}$), contributing 78% to eBC mass. The higher AAE fraction (AAE=2.04) is likely eBC from BB


(eBC$_{bb}$). Source receptor model calculations (FLEXPART) showed that Continental Europe and western
Russia were the main source regions both of elevated eBC$_{bb}$ and eBC$_{ff}$.

A relative increase in the OC fraction in PM$_{2.5}$ (+3.2% yr$^{-1}$) and PM$_{10}$ (+2.3% yr$^{-1}$) underscores

the importance of biogenic sources at Birkenes (BSOA and PBAP), which were higher in the vegetative
season and dominated both fine (53%) and coarse (78%) OC. Furthermore, 77–91% of OC in PM$_{2.5}$,
PM$_{10-2.5}$ and PM$_{10}$ was attributed to biogenic sources in summer vs. 22-37% in winter. The coarse
fraction had the highest share of biogenic sources regardless of season and was dominated by PBAP,
except in winter.

Our results show a shift in aerosol composition at Birkenes and thus also in the relative source

contributions. The need for diverse off-line and on-line carbonaceous aerosol speciation to understand
carbonaceous aerosol sources, including their seasonal, annual, and long-term variability has been
demonstrated.

## 1.  Introduction

Carbonaceous aerosol has been studied intensively over the last 20 years due to its influence on

radiative forcing (Bond et al., 2013; Myhre and Samset, 2015; Lund et al., 2018), both directly by
scattering and absorption of sunlight, and semi directly and indirectly by influencing cloud properties
(Boucher et al., 2013; Hodnebrog et al., 2014,; Myhre et al., 2013). It also contributes to the burden of
respiratory and cardiovascular disease (Janssen et al., 2012; WHO, 2013). Consequently, carbonaceous
aerosol [here: elemental carbon (EC) and organic carbon (OC)] is measured regularly in air monitoring
networks (e.g., Tørseth and Hov, 2003; Tørseth et al., 2012; UNECE, 2019; Hjellbrekke, 2020).
Carbonaceous aerosol has an atmospheric lifetime of days to a few weeks and is thus relevant for
atmospheric long-range transport. Accordingly, the European Monitoring and Evaluation Programme
(EMEP) included OC/EC measurements in 2004 after a pioneering measurement campaign at 12
European sites from 2002–2003 (Yttri et al., 2007a; Tørseth et al., 2012), showing that carbonaceous
aerosol was a major constituent of the ambient aerosol in the European rural background environment,
accounting for 9–37% (OM = organic matter) and 1–5% (EC) of PM$_{10}$, and that OM was more abundant
than sulfate (SO$_4^{2-}$) at sites reporting both variables (Yttri et al., 2007a). Similar conclusions were found
from another long-term campaign, CARBOSOL (Gelencsér et al., 2007; Pio et al., 2007), which
monitored atmospheric aerosol and its components for two years at six sites along a west-east transect
extending from the Azores, in the mid-Atlantic Ocean, to K-Kuszta (Hungary), in centra Europe.

There are numerous carbonaceous aerosol sources, both anthropogenic, e.g. emissions from

combustion of fossil fuel and biomass, and biogenic, e.g. vegetation emitted terpene/isoprene oxidation,
and primary biological aerosol particles (PBAP) from e.g. plants and fungus (Bauer et al., 2002;
Donahue et al., 2009; Hallquist et al., 2009; Fröhlich-Nowoisky et al., 2016).

Detailed source apportionment and quantification of carbonaceous aerosol is challenging due to

it numerous sources, the complexity of atmospheric formation and the vast number of organic





compounds associated with carbonaceous aerosols. A few studies have addressed carbonaceous aerosol
sources in the European rural background environment using source-specific organic tracers (Gelencsér
et al., 2007; Szidat et al., 2009; Genberg et al., 2011; Gilardoni et al., 2011; Yttri et al., 2011a,b). These
consistently show that residential wood burning dominates OC in winter, whereas BSOA is the major
source in summer. PBAP makes a significant contribution to $PM_{10}$ in the vegetative season in the Nordic
countries, second only to BSOA (Yttri et al., 2011a,b). Fossil fuel sources typically dominate EC
regardless of season but residential wood burning emissions can be equally important and occasionally
dominate in the heating season (Zotter et al., 2014; Yttri et al., 2019). On-line high time-resolution
measurements by aerosol mass spectrometer (AMS) and aerosol chemical speciation monitors (ACMS)
have become available in recent years, complementing off-line analysis of organic tracers. In the
comprehensive study by Crippa et al. (2014), including 15 European rural background sites and 2 urban
sites, covering winter, spring and fall, hydrocarbon-like organic aerosol (OA) (11±5%) and biomass
burning OA (12±5%) contributed almost equally to the total OA concentration. The vast majority was
however attributed to secondary sources; i.e., semi volatile oxygenated OA (34±11%) and low-volatility
oxygenated OA (50±16%). Secondary oxygenated OA (OOA) can be both anthropogenic and biogenic,
however Crippa et al. (2014) did not draw any conclusions on this. Results presented by Bougiatioti et
al. (2014) show how freshly emitted biomass burning OA can be transformed to more oxidized OOA
after just a short time in the atmosphere when subject to high temperatures and high solar radiation.

Over the last decades, European anthropogenic emissions of secondary inorganic aerosol precursors,

e.g. ammonia ($NH_3$) and nitrogen oxides ($NO_x$), and non-methane volatile organic compounds
(NMVOC) have stabilized, and those of sulfur dioxide ($SO_2$) significantly reduced, following
implementation of the Gothenburg Protocol (Reis et al., 2012; UNECE, 2013; Matthews et al., 2020).
The anthropogenic carbonaceous aerosol is not regulated by any binding international protocol, although
co-benefit is expected from the regulation of $NO_X$ and NMVOC, which act as precursors of secondary
organic aerosol (Hallquist et al., 2009). $PM_{2.5}$ was included in the revised version of the Gothenburg
protocol (UNECE, 2013) in 2012, which states that effort should be directed towards sources that also
emit black carbon (BC), which inevitably also will influence OC.

Residential wood burning is a major source of carbonaceous aerosol in circumpolar countries (e.g.

Yttri et al., 2014) and even considered the most important source in Norway, accounting for 48% (2017)
of $PM_{2.5}$ (Grythe et al., 2019). This region also regularly experiences major wild and agricultural fires
(e.g. Stohl et al., 2006 and 2007). A growing number of studies show that residential wood burning is
more widespread in continental Europe than previously assumed and that its contribution to the ambient
carbonaceous aerosol can be substantial (Sillanpää et al., 2006; Gelencsér et al., 2007; Puxbaum et al.,
2007; Lanz et al., 2010; Maenhaut et al., 2012; Genberg et al., 2013; Fuller et al., 2014; Yttri et al.,
2019) and even dominating (Szidat et al., 2007; Herich et al., 2014). Residential wood burning is a
decentralized source in Europe and combustion typically takes place in small units where the emissions
are emitted without after-treatment. An economic downturn in Greece compelled households to burn





firewood and waste material as fuel costs rose, increasing residential wood burning emissions in urban
areas by 30% (Saffari et al., 2013). Future increases in European wood burning emissions might occur
due to climate change mitigation policies supporting the use of renewable and biofuels (van der Gon et
al., 2015). Denier van der Gon et al. (2015) conclude that European emissions from residential wood
burning are significantly underestimated, thus it appears timely to address how ambient carbonaceous
aerosol -particularly from biomass burning -have developed over the last two decades.
Kahnert et al. (2004) and Tørseth et al. (2012) highlight the importance of long-term measurements
(> 10 years) of carbonaceous aerosol. The Birkenes Observatory in southern Norway holds the longest
time series of OC and EC in Europe, dating back to 2001, including measurements in both the $PM_{10}$ and
the $PM_{2.5}$ fractions. Downwind of major anthropogenic emission regions in Europe, the Birkenes
Observatory is well suited to monitor air pollution from Continental Europe.
Here we apply positive matrix factorization (PMF) to identify sources of carbonaceous aerosol at
the Birkenes Observatory. Measurements of complementary species accompany OC/EC monitoring,
allowing us to understand these sources, their contribution and variability at time scales from minutes
to decades: organic tracers for biomass burning (levoglucosan), PBAP (arabitol, mannitol, trehalose and
glucose) and BSOA (2-methyltetrols), as well as high time resolution equivalent black carbon resulting
from biomass ($eBC_{bb}$) and fossil ($eBC_{ff}$) fuel combustion, derived from multiwavelength aethalometer
measurements.

**2.   Methodology**
**2.1    Sampling site**
The Birkenes Observatory (58°23'N, 8°15'E, 219 m above sea level, asl) is an EMEP/GAW (Global
Atmospheric Watch) supersite in southern Norway (Figure 1). The observatory is in the Boreo-nemorale
zone with mixed coniferous and deciduous trees (65% of the land use near the site); the remainder being
meadows (10%), low intensity agricultural areas (10%), and freshwater lakes (15%). Close to the
Skagerrak coast (~20 km) and at low altitude, the observatory experiences a maritime climate with
relatively mild winters and moderately warm summers. The prevailing wind is westerly/south westerly.
Figure S 1 shows ambient temperature and precipitation (2001–2018) at Birkenes. The nearest city is
Kristiansand (population ~61 000) 25 km to the south/south-west.

**2.2    Measurements and procedures**
**2.2.1   Off-line filter measurements**
We collected OC/EC, organic tracers and PM mass filter samples using two low-volume samplers with
a $PM_{10}$ and a $PM_{2.5}$ inlet. Quartz fiber filters (Whatman QM-A; 47 mm in diameter) were pre-fired (850
°C; 3 h). We conditioned the filters [20 ± 1°C; 50 ± 5% RH (relative humidity)] for 48 h before and after
exposure and weighed them to obtain PM mass. We kept filters in petri slides and stored them at 4 °C
after weighing and before OC/EC analysis. After OC/EC analysis and prior to organic tracer analysis



the samples were stored at -18 °C. Two field blanks were assigned to each month of sampling and were
treated in exactly the same manner regarding preparation, handling, transport and storage as the exposed
filters, except that they were not inserted in the samplers. We collected one sample per sampler per week
(168 hours), except for 14 August 2002–17 September 2008, when two samples were collected per
sampler per week; at 24 h and 144 h intervals. The sampling inlets are 2 m above the Observatory roof,
5 m above the ground level (~226 m asl). The OC/EC and PM mass time series date back to February
2001 and organic tracers back to January 2008 (monosaccharide anhydrides) and January 2016 (sugars,
sugar-alcohols and 2-methyltetrols).

We performed thermal-optical analysis (TOA, Sunset Laboratory OC/EC instrument), using

transmission for charring correction. We used the Quartz temperature programme in 2001–2008 and
EUSAAR-2 (Cavalli et al., 2010) from 2008. We compare the two temperature programmes for $PM_{2.5}$
samples collected in 2014 in Supplementary Sect. S1. OC/EC instrument performance is regularly inter-
compared under the joint EMEP/ACTRIS quality assurance and quality control effort (e.g. Cavalli et
al., 2013).

Until 2014, we determined monosaccharide anhydrides (levoglucosan, mannosan, galactosan)

in $PM_{10}$ using high-performance liquid chromatography high-resolution time-of-flight mass
spectrometry (HPLC-HR-TOFMS) in negative electrospray ionization mode according to the method
of Dye and Yttri (2005). After 2014, we use ultra-performance liquid chromatography (UPLC), with
two Waters columns (2 x 2.1 x 150 mm HSS T3, 1.8 μm, Waters Inc.). Changing the column improved
the chromatographic resolution, allowing the analysis of sugars, sugar-alcohols and 2-methyltetrols. We
identified the monosaccharides anhydrides based on retention time and mass spectra (accurate mass and
isotope pattern) of authentic standards (Table S 1). Isotope-labelled standards of levoglucosan,
galactosan, arabitol, mannitol, trehalose and glucose were used as internal recovery standard (Table S

1).

Weekly OC/EC, $PM_{10}$, $PM_{2.5}$ are publicly available on EBAS (http://ebas.nilu.no). Mean values

(daily/weekly/seaonal/annual) used below, merging of data from the old and new Birkenes sites, and
quality assurance of the filter data are detailed in Sect. S1. We used the Mann-Kendall test (Mann, 1945;
Kendall, 1975; Gilbert, 1987) to identify significant trends in the filter based measurements, and the
Theil-Sen slope (Theil, 1958; Sen, 1968; Gilbert, 1987) to quantify the trends (Sect. S2).

**2.2.2   Online measurement and source apportionment of absorption coefficients**
We determined Absorption coefficients ($B_{Abs}$) using a multi-wavelength absorption photometer (AE33
Aethalometer, Magee Scientific). Here we performed source apportionment using the aethalometer
model (Sandradewi et al., 2008) to determine $eBC_{bb}/eBC_{ff}$. However, the aethalometer model requires
*a-priori* knowledge of the aerosol Ångstrøm exponents (AAE), uncertainties in which can lead to large
variation in the magnitude of the resulting time series and negative concentrations during some periods.
Often, the aethalometer model yields negative concentrations for any single input AAE pair. Therefore,





we also used a novel positive matrix factorization (PMF) application finding two factors, a low AAE
factor (0.9) and a higher AAE factor (2.04) identified as $eBC_{ff}$ and $eBC_{bb}$, respectively. Uncertainties
were assessed using bootstrapping ($n$=2000). The advantages of the PMF are that no *a-priori* knowledge
of the factor AAEs is required, no periods of negative concentration result, deviations from a strict
power-law dependence of $B_{Abs}$ on wavelength (e.g. due to degradation of light absorbing components in
the atmosphere or instrument errors/bias) are permitted, and poorly fitting data are assigned to a residual.
Meanwhile bootstrapping allows estimation of uncertainties, the methodology of the PMF analysis and
aethalometer data post-processing are detailed in Sect. S3 (Table S 2).

**2.3    FLEXPART model simulations**
We investigated the origin of the observed eBC with a Lagrangian transport model (FLEXPART v10.4,
Pisso et al., 2019). The model, powered by European Centre for Medium-Range Weather Forecasts with
137 vertical layers and a horizontal resolution of 0.1°×0.1° tracks simulated particles arriving at the
receptor 30 days backwards in time (retro-plume mode) and accounts for gravitational settling, dry and
wet deposition (Grythe et al., 2017), turbulence (Cassiani et al., 2014), unresolved mesoscale motions
(Stohl et al., 2005) and includes a deep convection scheme (Forster et al., 2007). Output consists of an
emission sensitivity (0.5°×0.5° resolution), a quantitative measure for the particle mass concentration at
the receptor resulting from a unit emission flux at the Earth's surface. The emission sensitivity can also
be interpreted as a probability distribution field of the particle's origin, used in the present study to
identify possible source regions of eBC.

**2.4    Positive Matrix Factorisation analysis on filter data**
We performed PMF (See Sect. S3 for a description of the analysis principal and S4 for its application to
filter data) using the following as input data: OC (in $PM_{2.5}$ and $PM_{10-2.5}$), EC (in $PM_{10}$), levoglucosan,
mannosan, galactosan, arabitol, mannitol, trehalose, glucose, V, Mn, Ti, Fe, Co, Ni, Cu, Zn, As, Cd, and
Pb (all in $PM_{10}$), $SO_4^2$, $NO_3^-$, $NH_4^+$, $Ca^{2+}$, $Mg^{2+}$, $K^+$, $Na^+$, $Cl^-$ (from open filter face). Table S 3 shows
miscellaneous settings of the PMF analysis of these data including missing data treatment and an
assessment of the PMF performance. The input data and error estimates were prepared using the
procedure suggested by Polissar et al. (1998) and Norris et al. (2014), see Sect. S3.

Source apportionment via PMF is based on the temporal variability of the components. It is

expected that significant contributions to carbonaceous aerosol at Birkenes is via long-range
atmospheric transport (LRT), alongside more local sources. Local and LRT sources will have different
temporal variability, and significant mixing of air masses and chemical transformation is expected for
the latter, i.e., factor profiles at Birkenes are expected to differ somewhat from emission profiles at the
source, even though the profile is distinctive enough for source attribution. Because of this we did not
attempt to constrain factor profiles via e.g. ME-2 (Canonaco et al., 2013) since Birkenes, as a relatively
clean rural background site, is unlikely to receive unprocessed emissions. Furthermore, mixed


contributions to a factor can in some cases be resolved *a posteriori* for source quantification (i.e. if it is
clear where mass should be reassigned), without potentially perturbing the output factor time series.

### 2.4.1    Identification of PMF factors

The BB factor appears well confined in the PMF solution (Figure 2, Table S 4), explaining all the
monosaccharide anhydrides (95–98%). $OC_{BB}$ was almost exclusively (87%) in the fine fraction of $PM_{10}$.
Other key qualifiers derived from the BB factor are the ratios listed in Table 1, which are highly
comparable to the results obtained by $^{14}C$-analysis reported in the comprehensive study by Zotter et al.
(2014). The BB factor is elevated in the heating season and peaks in winter, pointing to residential
heating as the major source.

The TRA/IND factor explained most EC (50%), the majority of the trace elements Pb (84%),
Zn (82%), Cd (81%), As (78%), V (70%), Ni (69%), Cu (62%) and Co (42%) and a noticeable fraction
of $SO_4^{2-}$ (20%), which suggests influence of various anthropogenic emissions. IND/TRA explained a
small fraction of fine OC (10%) and a negligible fraction of coarse OC (4%). The majority of OC (88%)
resides in the fine fraction, which is in line with its combustion-derived origin. The high EC fraction
unambiguously points to combustion processes, and the low OC/EC ratio (1.4 for $PM_{2.5}$) towards a
substantial, but not exclusive, influence from vehicular traffic. Cu and Zn result from brake wear (Fomba
et al., 2018), whereas tire wear is an additional source of Zn (Pacyna et al., 1996), corroborating the
influence of vehicular traffic to the TRA/IND factor. Ni and V are commonly associated with
combustion of heavy oil (Viana et al., 2008), As, Cd and Pb with combustion of coal, and to a lesser
extent oil, but also from metallurgic activity (Pacyna et al., 1986). The TRA/IND factor has a minimum
in summer and shows minor variability for the rest of the year. A similar drop in the vehicular traffic
factor in summer for Helsinki was shown by Saarikoski et al. (2008).

The PMF analysis confined the majority of coarse OC (53%) and essentially all (82–93%) of
the PBAP tracers (arabitol, mannitol, trehalose, and glucose) within one factor (PBAP). The PBAP
factor has a pronounced seasonal variability with increased levels in the vegetative season and nearly
absent outside of it, as previously described for coarse OC (Yttri et al., 2007a) and PBAP tracers (Yttri
et al., 2007b) at Birkenes.

2-methyltetrols (92–96%) are oxidation products of isoprene (Claeys et al., 2004) and are almost
exclusively attributed to the $BSOA_{SRT}$ (SRT=Short Range Transport) factor, which explains 9% of fine
OC and 13% of coarse OC. The complete absence of EC and the presence of $SO_4^{2-}$ (17%) underpins the
secondary nature of this factor, which is present in summer with tail ends in late spring an early fall. The
$BSOA_{SRT}$ time series increases abruptly in the transition May/June, as leaves unfold, and subsides
equally rapid in the beginning of October when trees shed their leaves. The near absence of 2-
methyltetrols prior to May/June suggests that the 0.5–1.5 months earlier onset of the vegetative season
in Continental Europe (Rötzer and Chmielewski, 2001) is not reflected by the 2-methyltetrols
observations at Birkenes, indicating a short atmospheric lifetime for 2-methyltetrols. Consequently,





local isoprene emissions likely explain the observed concentrations of 2-methyltetrols at Birkenes,
questioning to what extent the $BSOA_{SRT}$ factor includes a continental BSOA contribution. Similar
sources (deciduous and coniferous trees), temperature dependent emissions, and formation rates, suggest
that particulate phase oxidation products of mono- and sesquiterpenes are accounted for by the isoprene-
derived $BSOA_{SRT}$-factor as well, but with a similar issue concerning local versus LRT contribution, as
proposed for the 2-methyltetrols.

The MIN factor is defined by its content of Ti (93% of total), Fe (75%), Mn (52%) and $Ca^{2+}$
(39%) (Figure 2, Table S 4), well-known constituents of mineral dust (e.g. Alastuey et al., 2016). It also
contains some of the elements that dominate the TRA/IND factor, including Co (43%), Cu (20%), Ni
(17%) and V (14%), indicating anthropogenic influence. Notably, 31% of fine OC is attributed to the
MIN factor, whereas it explains 13% of coarse OC. This corresponds to that reported by Kyllönen et al.
(2020) for the Subarctic site Pallas (Finland) where 29% of the fine OC was apportioned to the mineral
dust factor. Waked et al. (2014) found a similar result for Lens (France) where the mineral dust factor
explained 15% of OC. No information on the size distribution was available in Kyllönen et al. (2020)
and Waked et al. (2014), whereas in the present study 86% of OC in the MIN factor resides in the fine
fraction of $PM_{10}$. Since mineral dust typically resides in the coarse fraction of $PM_{10}$ (Ripoll et al., 2015),
one would expect the same for its carbon content, e.g. as $CaCO_3$. More efficient deposition of coarse
mode mineral dust during LRT is one possible explanation but mixing of air masses is more likely, as
13% of the EC also resides in this factor. The high OC/EC ratio in the unweighted MIN factor profile
(18 for $PM_{2.5}$) indicates a minor primary combustion particle influence, and the absence of levoglucosan
shows that the EC content originates from fossil fuel combustion (consistent with some TRA/IND
influence). Using Eq. (1), 8% of the MIN factor's fine OC content is attributed to combustion of fossil
fuel OC ($OC_{PrimFF}$), whereas the corresponding percentage for $PM_{10}$ OC is 7%. If all $Ca^{2+}$ and $Mg^{2+}$ in
the MIN factor was present as either Calcite ($CaCO_3$) or Dolomite $CaMg(CO_3)_2$, the $CO_3^{2-}$-carbon would
account for no more than 3% of the factor's $PM_{10}$ OC content, and 22% if all reside in its coarse fraction.
This shows that the OC content of the MIN factor mostly originates from other sources than mineral
dust and combustion of fossil fuel. The MIN factor is most abundant in spring and early summer, as
seen by Waked et al. (2014), and is associated with southern air masses, as seen for the dry and warm
period in the transition of May/June 2018 (Figure 3) when there was a pronounced peak in the MIN
factor time series (Figure 2). Indeed, the mean ambient temperature was 4˚C higher in May 2018 than
for May 2001–2018, whereas it was 2.4˚C higher for June 2018 than for June 2001–2018. We thus
suggest that the climatological conditions that activate mineral dust sources also favours BSOA
formation and that the majority of both fine (92%) and coarse fraction (78%) OC in the MIN factor is
LRT BSOA ($OC_{BSOA,LRT}$).

$$OC_{Fossil,primary,MIN} = [EC_{MIN}] \, x \left(\frac{OC}{EC}\right)_{TRA/IND}, \left(\frac{OC}{EC}\right)_{TRA/IND} = 1.4 \qquad\qquad Eq.\,1$$



The majority of $NH_4^+$ (77%) and $NO_3^-$ (68%) reside in the $NH_4NO_3$ factor, which points to
secondary inorganic aerosol (SIA) formation during LRT. This is supported by a noticeable contribution
of $SO_4^{2-}$ (35%) to the $NH_4NO_3$ factor, as well. The factors content of $NO_2$ (30%) points towards a
combustion-derived origin of $NO_3$, as does EC (13%). The factor's OC content is comparable to that
seen for the BB factor. The factor is most pronounced in winter and spring.
The SS factor was recognized by its high $Cl^-$ (96%), $Na^+$ (87%) and $Mg^{2+}$ (79%) fractions. The
$K^+/Na^+$ (0.036), $Ca^{2+}/Na^+$ (0.034) and $SO_4^{2-}/Na^+$ (0.282) ratios derived from the SS factor closely
resembles these ratios in sea water (0.037, 0.038 and 0.252) (Stumm and Morgan, 1995), further
demonstrating the successful separation of this factor.

**3.  Results and discussion**
**3.1     Levels and trends of carbonaceous aerosol and organic tracers**
Annual mean carbonaceous aerosol concentrations at Birkenes (2001–2018) are among the lowest in
Europe (Yttri et al., 2007a; Yttri et al., 2019), with OC from 0.56–1.07 µg C m$^{-3}$ for $PM_{10}$ and 0.50–
0.93 µg C m$^{-3}$ for $PM_{2.5}$, and EC from 0.05–0.15 µg C m$^{-3}$ (Figure 4; Table S 4). EC, being from
combustion that generates fine PM, was almost exclusively associated with $PM_{2.5}$, whereas OC was
abundant also in the coarse fraction ($PM_{10-2.5}$), particularly in summer and fall (Figure 4). The
correlation between OC and EC varied by season (Table S 6) and was highest in the heating season,
reflecting the contribution of biogenic, non-EC sources, such as BSOA and PBAP in the vegetative
season. The higher $R^2$-values for $PM_{2.5}$ compared to $PM_{10}$ can partly be attributed to PBAP, which
mainly resides in $PM_{10-2.5}$.
The variability of the annual mean OC (15–22%) and EC (27%) concentrations was comparable
to the major secondary inorganic aerosol (SIA) ($SO_4^{2-}$, $NO_3^-$, $NH_4^+$) and sea salt (SS) aerosol species
($Na^+$, $Mg^{2+}$, $Cl^-$) (25–31%). A difference of > 60% between consecutive years was observed for OC and
EC in $PM_{10}$ and $PM_{2.5}$, whereas 160% was seen for OC in $PM_{10-2.5}$. It is important to note that despite
decades of $SO_2$, $NH_3$ and $NO_x$ mitigation efforts, SIA dominates $PM_{10}$ mass (29–52%) most years,
followed by carbonaceous aerosol (24–40%) and SS aerosol (10–28%) (Figure 5; Table S 7). SIA
constituents were also the largest $PM_{10}$ fraction during air pollution episodes (Table S 8), reflecting that
Birkenes is affected by major SIA precursor emission regions in Continental Europe.
Levels of total carbon (TC) and PM fractions are shown in Table S 9 and Table S 10, respectively
for completeness. In the following sections we discuss the OC and EC fractions separately in detail.

**3.1.1     Organic carbon**
We found no significant trend for OC in $PM_{10}$ ($OC_{PM10}$). For fine OC in $PM_{2.5}$ ($OC_{PM2.5}$) there was a
minor decrease (-0.8% yr$^{-1}$), whereas there was a minor increase for coarse OC ($OC_{PM10-2.5}$) (0.8% yr$^{-1}$)
(Table S 11). The anthropogenic fraction of OC observed at Birkenes likely has a downward trend as
found for EC, (Sect. 3.1.2) but the substantial influence of natural sources demonstrated in the present,


as well as in previous, studies (Yttri et al., 2011b), explains the general lack of trends for OC.
The OC time series are characterized by two years where the annual mean was substantially
higher (2006) and lower (2012) than the proceeding and the following year (Figure 4). The increased
level in 2006 was most pronounced in the fine fraction and in all seasons except spring, whereas the
drop in 2012 mainly was attributed to the coarse fraction and was observed in all seasons. The $OC_{PM10-2.5}$
annual mean time series is characterised by a stepwise increase from 2001 up to, and including, 2006,
after which the concentration dropped and showed minor annual variability, except for the very low
annual mean of 2012. After 2015, there are indications of a similar stepwise increase as seen for 2001–

343 2006.

The $OC_{PM10-2.5}$ contribution to $OC_{PM10}$ ranged from 18–35% on an annual basis (2001 excluded due to
data capture <50%), and levels were highest in summer and fall. Previous studies (Simpson et al.,
2007; Yttri et al., 2011a,b) showed that BSOA largely dominates the fine carbonaceous aerosol in
summer at Birkenes, whereas the present study shows that Birkenes regularly experiences major air
pollution events in spring, as a result of long-range atmospheric transport (LRT) (Table S 4, Table S 7
and Table S 8. Hence, both biogenic sources and LRT explain the observed seasonality of fine OC.
We attribute elevated $OC_{PM2.5}$ in winter 2010 to residential wood burning emissions as discussed
in Sect. 3.2.1. Only on five occasions did the seasonal mean of $OC_{PM2.5}$ exceed 1 µg C m$^{-3}$, four of those
in the first three years of the time series. The highest mean was observed in summer 2002 (1.4 µg C m$^{-3}$)
when wildfires in Eastern Europe influenced Birkenes (Yttri et al., 2007a). The four other occasions,
spring (2001, 2002, 2003 and 2018), also saw prolonged episodes of PM air pollution with the hallmark
of LRT; i.e., elevated $SO_4^{2-}$, $NO_3^-$ and $NH_4^+$. According to our PMF analysis (See Sect. 3.2) there are
several anthropogenic and biogenic sources likely to contribute to fine OC at Birkenes, whereas coarse
fraction OC is dominated by a single source, PBAP (Yttri et al., 2007 a,b; Yttri et al., 2011 a,b; Glasius
et al., 2018). Hence, it is not surprising that $OC_{PM2.5}$ was the dominant OC fraction, accounting for 70–
89% $OC_{PM10}$ on an annual basis.

### 361 3.1.2 Elemental carbon

Notably, EC levels dropped from 2007–2008, contrasting with the annual mean OC time series,
(Figure 4 and Table S 4). This major downward trend of EC clearly points to changing source
contributions to EC at Birkenes. We rarely observed seasonal means exceeding 0.15 µg C m$^{-3}$; only in
winter 2006, 2007 and 2010, spring 2001, 2003 and 2007, and fall 2005 and 2011. Weekly samples
exceeded 0.5 µg C m$^{-3}$ for three samples only, all associated with LRT.
A statistically significant reduction was calculated for EC in $PM_{10}$ (-3.9% yr$^{-1}$) and $PM_{2.5}$ (-4.2%
yr$^{-1}$) (Table S 11), corresponding well with $SO_4^{2-}$ (-3.8% yr$^{-1}$) and $PM_{2.5}$ (-4.0% yr$^{-1}$). The trend for EC
was most pronounced in spring and summer (-4.0 – -5.9% yr$^{-1}$) (Table S 12), as seen for $SO_4^{2-}$ (-4.2 – -
6.4% yr$^{-1}$) and $PM_{2.5}$ (-3.0 – -4.4% yr$^{-1}$) (Table S 12). The EMEP model finds a somewhat lower
reduction for EC (-3.0 % yr$^{-1}$) for 2001–2017 (EEA, 2020) with the largest emission reductions for the



road transport (83 kt; -3.6% yr$^{-1}$) and off-road categories (44 kt; -3.7 % yr$^{-1}$) (https://www.ceip.at), which are sectors with a minor seasonal variability. We suggest that these sectors explain the downward trend observed for EC at Birkenes, and that the seasonality of the EC trend is due to the substantial contribution from less abated sources, such as domestic heating in winter and fall. Notably, modelled EC emissions are unchanged for the category other stationary combustion for 2001–2016 (-1 kt; -0.08% yr$^{-1}$) (https://www.ceip.at), which includes residential heating, and wood burning in particular.

Effective abatement of SIA precursors and fossil EC, along with a high natural source contribution to OC, largely explains why the OC fraction increased significantly for PM$_{2.5}$ (+3.2% yr$^{-1}$) and PM$_{10}$ (+2.4% yr$^{-1}$), whereas it decreased for the EC fraction (-3.9 – -4.5% yr$^{-1}$) (Table S 13). The largest increase (OC) and decrease (EC) was seen in the vegetative season (Table S 14) when BSOA and PBAP increase and the influence of poorly abated sources such as domestic heating is low. Consequently, these results demonstrate a long-term change in the aerosol chemical composition at Birkenes and thus also in the relative source composition of PM.

### 3.1.3 Levoglucosan

Levels of levoglucosan and other organic tracers are given in Table S 15, whereas other organic tracers (arabitol, mannitol, trehalose, glucose, and 2-methyltetrols) are discussed in Sect. S6.

The statistically significant decrease of levoglucosan (-2.8% yr$^{-1}$) at Birkenes for 2008–2018 (Figure 6; Table S 11), and the fact that biomass burning levels observed at Birkenes are largely explained by continental emissions (Figure 7) might indicate that wood burning emissions in continental Europe are declining. However, surprisingly, we find no significant trend for levoglucosan on a seasonal basis (Table S 12). Furthermore, and although one should be careful drawing conclusions from non-significant outcomes, it is worth noting that the levoglucosan to EC ratio most likely increased (+2.8% yr$^{-1}$, CI = -3.5 – +6.5% yr$^{-1}$ and +2.3% yr$^{-1}$, CI = -2.2 – 5.0 % yr$^{-1}$) for the period 2008–2018, whereas it most likely decreased (-1.8, CI = -10.6 – +1.8 and -3.6% yr$^{-1}$, CI = -9.8 – +1.3% yr$^{-1}$) for the levoglucosan to OC ratio (Table S 13). A more efficient abatement of fossil sources than biomass burning would explain the levoglucosan to EC increase, whereas we fail to see a similar trend for the levoglucosan to OC ratio, as prevailing natural sources mask the assumed reduction in fossil OC of anthropogenic origin.

The levoglucosan time-series provides a hitherto unprecedented opportunity to validate European residential wood burning emission inventories at a decadal time basis. Unfortunately, the inventories suffer from non-harmonized emission reporting and lack of condensable organics (van der Gon et al., 2015, Simpson et al., 2019), which hampers any reliable attempt for such validation. Given the uncertainties in the trend calculations (i.e. annual vs. seasonal trends), more work is needed to investigate trends in levoglucosan and biomass burning, foremost by continuation of the actual time series. Such efforts should be initiated immediately given the numerous studies that point to residential wood burning as a major source of air pollution in Europe (e.g. Denier van der Gon et al., 2015; Yttri et al., 2019).



### 3.2 Sources of carbonaceous aerosol at Birkenes

We used PMF to apportion carbonaceous aerosol at Birkenes for 2016–2018. The time period was restricted by organic tracer data availability. Carbonaceous aerosol annual means for 2016–2018 were within the long-term annual mean (±SD) for OC, and only slightly lower for EC in 2016 and 2017 and are thus representative of the longer time series. Six out of seven factors identified in contribution-weighted relative profiles from PMF (Figure 2; Table S 4) were associated with significant amounts of carbonaceous aerosol. This includes factors for mineral dust-dominated (MIN), which OC content is associated mainly with LRT BSOA (BSOA$_{LRT}$), traffic/industrial-like (TRA/IND), biogenic secondary organic aerosol (BSOA$_{SRT}$), which is short-range transported, primary biological aerosol particles (PBAP), biomass burning (BB), and ammonium nitrate dominated (NH$_4$NO$_3$). The sea salt aerosol factor (SS) had a negligible (<1%) carbonaceous aerosol content.

The MIN factor (31%) explained the largest fraction of fine OC, whereas BB (17%), NH$_4$NO$_3$ (17%) and PBAP (16%) had almost equally large shares, as did IND/TRA (10%) and BSOA$_{SRT}$ (9%) (Figure 8). Coarse OC was by far most abundant in the PBAP factor (53%), whereas BSOA$_{SRT}$ (13%), MIN (13%) and NH$_4$NO$_3$ (12%) explained almost equally large shares. For the other factors, coarse OC was minor. EC was apportioned to only five factors of which TRA/IND (50%) dominated by far. BB made a 21% contribution and MIN and NH$_4$NO$_3$ equally large shares (13%). The 3% apportioned to PBAP is an assumed analytical artefact (See Sect. 3.2.2 for details).

The BB, NH$_4$NO$_3$ and TRA/IND factors are considered entirely anthropogenic, BSOA$_{SRT}$ and PBAP exclusively natural, whereas MIN is mixed (Figure 8). Natural (54%) and anthropogenic (46%) sources contributed almost equally to fine OC (Figure 8) annually, so also in spring and fall (51% natural), whereas natural sources prevailed in summer (77%) and anthropogenic in winter (78%). Natural sources dominated coarse OC annually (78%) and in all seasons (70–91%), except winter (37%). We consider the minor fraction of coarse OC attributed to carbonate-carbon (3%) to be of natural origin. The findings for OC in PM$_{10}$ are rather like that of PM$_{2.5}$, only that the natural contribution is somewhat more pronounced due to the influence from a mostly naturally influenced coarse OC fraction.

#### 3.2.1 Anthropogenic carbonaceous aerosol sources

According to PMF, BB accounted for 14–17% of OC annually, considering both PM$_{2.5}$ and PM$_{10}$ vs. only 6% of coarse OC. BB was by far the major contributor to OC in winter (35–37%) and by far the most minor contributor in summer (2–3%) (not considering SS). Spring and fall are transition seasons where BB still made a substantial 14–19% contribution to OC. BB explained 22% of EC annually (excluding EC$_{PBAP}$, which we assume is an analytical artefact, see Sect. 3.2.2), hence fossil fuel combustion (78%) was the major source. Emissions from residential wood burning increased in the heating season but fossil fuel sources dominated EC even in winter (66%). It cannot be excluded that part of levoglucosan originates from wildfires in summer, spring, and fall, though this itself may be due



to anthropogenic activity. However, the levoglucosan/mannosan (L/M) ratio indicates minor variability
in the source composition throughout the year (See Sect. S5), suggesting one dominating source.
The 78%:22% split of EC into fossil fuel combustion and biomass burning derived from PMF is
supported by high time resolved concentrations of $eBC_{BB}$ and $eBC_{FF}$ derived from multiwavelength
aethalometer measurements of the absorption coefficient, following the PMF-approach of Platt et al.
(in prep.). With this approach we find $eBC_{BB}/eBC_{TOT}$=28% (Table 2). Meanwhile, using the
aethalometer model and $AAE_{FF}$=0.9 and $AAE_{BB}$=1.68 (Zotter et al. 2017) as input we find
$eBC_{BB}/eBC_{TOT}$=48%, however the aethalometer model is extremely sensitive to the input AAE and the
AAE values suggested by Zotter et al. (2017) are only recommended where no *a priori* information on
the AAEs is available and a significant advantage of the PMF approach by Platt et al., (in prep.) is that
the AAE is an output.

Source regions of elevated (70[th] percentile) and low (30[th]) winter and summertime $eBC_{BB}$ (and

$eBC_{FF}$) observed at Birkenes for 2018 were studied using the approach of Hirdman et al. (2010). The
results show that Birkenes is a receptor of LRT exclusively from Continental Europe for elevated $eBC_{BB}$
and $eBC_{FF}$ levels (Figure 7), both in summer and winter. This is consistent with a lack of diurnal
variation in either $eBC_{BB}$ or $eBC_{FF}$, likely because there are few local sources at Birkenes. The main
source regions extend from the Atlantic coast in the west to the Ural Mountains in winter, whereas the
regions in summer are confined to Eastern Europe and western Russia (but not as far east as the Urals).
Notably, the Nordic countries do not contribute to elevated levels except for southern parts of Finland
in summer. The footprints are almost identical for $eBC_{BB}$ and $eBC_{FF}$ both for summer and winter. High
similarity in winter is not a surprise, as the footprint covers such a wide area and because wood burning
for residential heating is common in several European countries. The summertime footprint is a
subsection of the wintertime footprint that covers an area well-known for severe wildfires and
agricultural fires (Stohl et al., 2007 and Yttri et al., 2007a), and thus agrees with previous studies.
Further, Sciare et al. (2008) point to the European countries bordering the Black Sea as having high
carbonaceous aerosol of fossil origin. Low $eBC_{BB}$ and $eBC_{FF}$ levels at Birkenes are consistent with
airmasses that have an oceanic or terrestrial origin at high latitudes, mainly from the Arctic. Notably,
the 30% highest values explain 74% of $eBC_{BB}$ at Birkenes for the actual period, hence LRT is decisive
not only for episodes of high concentrations but also largely explains the mean concentration. All $eBC_{BB}$
and $eBC_{FF}$ observations included in the 70[th] percentile was made in winter despite the less pronounced
seasonality of $eBC_{FF}$ compared to $eBC_{BB}$.

To generate a longer BB time series of $OC_{BB}$ and $EC_{BB}$ we combine the levoglucosan time series

(2008–2018) with levoglucosan/OC and levoglucosan/EC ratios derived from the BB factor of the PMF
analysis (Table 1; See Sect. S 5 for details). Depletion of levoglucosan by OH oxidation is more likely
in summer (Hoffmann et al., 2010; Yttri et al., 2014), still we assume that levels mostly reflect biomass
burning emissions in all seasons.
$EC_{BB}$ levels were elevated in the heating season (Figure 9; Table S 16). A strong temperature
influence is illustrated by a 9°C difference in the 25th percentile of wintertime temperatures in 2015 (-
0.3°C) and 2010 (-9.3°C) (Figure S 1), which experienced the lowest (19 ng m$^{-3}$) and the highest (84 ng
m$^{-3}$) winter-time mean concentration of $EC_{BB}$, respectively. Winter 2010 was exceptionally cold due to
a negative North Atlantic Oscillation, and the only occasion when $EC_{BB}$ exceeded $EC_{FF}$, with annual
mean $EC_{BB}$>60% higher than the long-term mean. Pronounced interannual variability was seen for the
wood burning contribution in winter, from 21–60% to EC, with the lowest fractions occasionally
matched by those in spring and fall, typically ranging between 20–30%. $EC_{BB}$/EC was small in summer
(4–15%), considerably less than other seasons, except in 2008, where we calculate a substantial 30–40%
contribution. Levoglucosan cannot be used to differentiate emissions from residential wood burning,
wildfires and agricultural fires; exceptions are major wildfire and agricultural fire episodes identifiable
by unusual high concentrations and traced by source receptor models/satellite data for plumes/burnt
areas (Yttri et al., 2007a, Stohl et al., 2007). Influence from major wildfires in Eastern Europe caused a
summertime peak in fine OC and EC in 2002 at Birkenes (Yttri et al., 2007a). In June 2008, the largest
wildfire in Norway since the Second World War raged 25 km northeast of the Birkenes Observatory,
with an area of 30 km$^2$ burnt. The observatory was downwind of the fire on only one day, according to
FLEXPART (Figure S 2). Despite this, the levoglucosan concentration for the weekly filter sample was
153 ng m$^{-3}$, by far the highest in one decade of sampling. Notably, the annual mean concentration of
levoglucosan for 2008 increased by nearly 35% and $EC_{bb}$ contributed significantly to EC for summer

2008.

The seasonality of $OC_{BB}$ (Figure 9) was like $EC_{bb}$. Mean wintertime $OC_{BB}$/OC was 39–40% and
>50% in 2010 and 2012, considering both $PM_{10}$ and $PM_{2.5}$. The summertime contribution was typically
<5%, reflecting both low levoglucosan levels and major influences from BSOA and PBAP, which peak
in summer. Notably, five of the seven highest weekly OC concentrations for the $PM_{10}$ time-series were
attributed to emissions from major wildfires in Eastern Europe, i.e., August 2002, and May/September
2006, and thus prior to the initiation of the levoglucosan time series. The local wildfire episode in
summer 2008 caused a substantial increase in $OC_{bb}$/OC (13–18%), which is within the lower range of
that observed for spring (12–27%) and fall (13–39%).

### 3.2.2   Biogenic carbonaceous aerosol sources

The general lack of PBAP tracers in the MIN (<1%) and SS (<2%) factors and no sea salt and Ti in the
PBAP factor, implies that soil and sea spray aerosol do not contribute to PBAP at Birkenes, although
this has been shown elsewhere (O'Dowd et al., 2004; Jia and Fraser, 2011). PBAP represented by
glucose, arabitol and mannitol appears to be associated with leaves rather than soil material and to be a
source of local origin (Samaké et al., 2019). However, even large PBAP, such as birch pollen (avg. diam.
22 μm), has a potential for long range atmospheric transport of 1000 km due to its low density,
hydrophobic nature, release during favourable dispersion conditions, and (often) emission height > 10



m (e.g. Sofiev et al., 2006; Skjøth et al., 2007).
The PBAP factor concentration was nearly one order of magnitude higher in summer and fall
than in winter and was the major contributor to coarse OC for all seasons except winter, particularly in
summer (54%) and fall (69%) (Figure 8). These are conservative estimates, as 3–9% of the PBAP tracers
reside in the $BSOA_{SRT}$ factor, likely due to co-variability, as there is no scientific evidence linking
biologically formed sugars and sugar-alcohols to abiotic formation of BSOA. Notably, the PBAP factor
explained 20–26% of fine OC in summer and fall, being the major contributor in fall. Consequently,
PBAP was the major contributor to OC even in $PM_{10}$ in summer (31%) and fall (40%). The PBAP factor
even explained 16% of fine OC (Figure 8) annually, corresponding to 0.084 µg C m$^{-3}$, which is
marginally lower than the factor's content of coarse OC (0.113 µg C m$^{-3}$). Combined, this made PBAP
the most abundant contributor to OC in $PM_{10}$ along with the MIN factor (both 26%). Some PBAP tracers
partly reside in the fine mode (Carvalho et al., 2003); Yttri et al., 2007b) but the 43% $OC_{PBAP}$ found in
the fine fraction in the present study is higher than what has previously been reported for the actual
PBAP tracers at Birkenes; i.e. 6–7% (arabitol and mannitol), 20% (trehalose), and 33% (glucose) (Yttri
et al., 2007b). It cannot be excluded that the PBAP factor contains some fine OC from other sources e.g.
due to condensation, but although there is a seasonal co-variability with the $BSOA_{SRT}$ factor, only 2–
3% of the 2-methyltetrols were explained by the PBAP factor and there was a low correlation between
the PBAP and the $BSOA_{SRT}$, which questions this hypothesis.
Arabitol and mannitol are well-known tracers of fungal spores (Bauer et al., 2008), one of the
most abundant sources of PBAP (Elbert et al., 2007). Applying an OC to mannitol ratio of 5.2–10.8 for
fungal spores (Bauer et al., 2008; Yttri et al., 2011a), we estimate that 11–22% of $OC_{PBAP}$ (in $PM_{10}$)
comes from this source. Glucose is one of the primary molecular energy sources for plants and animals,
a building block of natural dimers and polymers (e.g. sucrose and cellulose), and thus ubiquitous in
nature and considered a PBAP tracer of general character, and clearly important for allocation of carbon
mass to PBAP. Nevertheless, a wider range of organic tracers ought to be tested in future PMF studies
to explore the potential of further separation of the highly heterogenic PBAP source, including cellulose,
but also amino acids. A greater diversity of PBAP tracers may also provide a more correct PBAP
estimate. The PMF approach used in the present study gives a somewhat higher, but overlapping,
estimate of $OC_{PBAP}$ at Birkenes for August 2016–2018 than Latin Hypercube sampling (LHS) for August
2009 (Yttri et al., 2011b) (Table 3). The LHS approach was based on *a priori* emission ratios, with
uncertainty ranges estimated in a similar way to a Monte Carlo analysis (though less computationally
extensive), and considered only the sum of fungal spores and plant debris as $OC_{PBAP}$, based on mannitol
(fungal spores) and cellulose (plant debris), whereas the PMF approach may pick other contributing, i.e.
co-varying, sources.
The 3% EC in the PBAP factor is substantially less than the 16% reported by Waked et al.
(2014), which stated that atmospheric mixing, PMF limitations and artifacts caused by thermal-optical
analysis could be plausible explanations. In the present study, low levels of coarse fraction EC



occasionally appear in summer and fall (Table S 5), following the seasonality of PBAP. This finding
does not exclude any of the three possibilities proposed by Waked et al., (2014), but supports the
suggestion by Dusek et al. (2017) that PBAP, or at least some types of PBAP, chars and evolves as
modern carbon EC during thermal-optical analysis. If $EC_{PBAP}$ indeed is an analytical artefact, then
constraining the PBAP factor to contain no EC, as suggested by Weber et al. (2019), should be done
with caution, as it will wrongfully apportion pyrolytic carbon generated from PBAP as EC to another
source. Thus, $EC_{PBAP}$ should rather be interpreted as $OC_{PBAP}$. With no $EC_{SS}$, no $EC_{BSOA,SRT}$ and $EC_{PBAP}$
an assumed analytical artefact, EC can be apportioned into a fossil fuel category ($EC_{FF}$), consisting of
the MIN, IND/TRA, and $NH_4NO_3$ factors (explains 0.2% of levoglucosan), and a non-fossil biomass
burning category ($EC_{BB}$), the BB factor. Some EC has been reported from meat cooking (Rogge et al.,
1991), which is a non-fossil source, but its influence is minor at Birkenes, as it has not been observed
based on concurrent ACSM-measurements and is not accounted for by levoglucosan.
Our PMF results support the use of $OC_{PM10-2.5}$ as a proxy of $OC_{PBAP}$, which has a pronounced
seasonality (Figure 4) with the highest seasonal mean concentration observed in summer for 15 of the
studied years and in fall for the three others (Table S 5). The seasonal mean exceeded 0.5 µg C m$^{-3}$ on
two occasions only; fall 2005 and fall 2006. With a few exceptions, $OC_{PM10-2.5}$ contributed more than
30% to $OC_{PM10}$ in summer and fall. The highest relative contribution (45–50%) to $OC_{PM10}$ were
exclusively observed in fall (2004, 2005, 2006, 2008, 2014, 2017), likely reflecting a combination of
high $OC_{PM10-2.5}$ concentrations and fine fraction $OC_{BSOA}$ declining at this time of the year. $OC_{PM10-2.5}$
made a substantially lower contribution to $OC_{PM10}$ in winter (mean: 13%) and in spring (mean: 19%)
compared to summer and fall, although contributions exceeding 25% were observed in spring for certain
years. Notably however, the PBAP factor explains 16% of fine OC, which would not be accounted for
using coarse OC as a proxy of $OC_{PBAP}$.
These numbers suggest that PBAP is a major, continuous contributor to OC in $PM_{10}$ at Birkenes
for a period of nearly two decades, and that it largely explains the seasonality. Estimates of PBAP levels
in the continental European rural background environment are largely lacking and should be undertaken
to explore PBAPs potential importance. With a longer vegetative season and a different climate, the
PBAP flux might be larger in more southerly countries, although the relative contribution might be lower
due higher overall OC levels. Waked et al. (2014) found that $OC_{PBAP}$ accounted for 17% of OC in $PM_{10}$
on an annual basis for an urban background site in Lens (Northern France), and between 5–6% in
winter/spring and 27–37% in summer/fall using PMF for source apportionment. These fractions are
comparable to those observed in the present study, albeit concentrations calculated by Waked et al.
(2014) were higher.
PBAP is a large OC source not included in many models. OC model closure, both for overall
levels and seasonality, would thus likely be improved in many cases by its inclusion. This appears to be
particularly important for regions with low anthropogenic influence. Birkenes is situated in the Boreo-
nemorale zone, a transition zone of the Nemorale and the Boreal zone, hence, findings made for this site





likely gives an indication of what can be expected for this scarcely populated, circumpolar region, which
by far is the largest terrestrial biome of the Northern Hemisphere. Hence, measurements in unperturbed
areas should include PBAP for a better understanding of background conditions. In turn, such
measurements may improve e.g. climate models; i.e., the aerosol climate effect under relatively clean
conditions.

Modelled estimates suggest a 10–40% contribution of BSOA to fine OC annually at Birkenes

(Simpson et al., 2007; Bergström et al., 2012). Hence, the 9% contribution of $OC_{BSOA,SRT}$ to fine OC,
and the 13% contribution to coarse OC (10% to OC in $PM_{10}$) found in the present study by PMF, appears
to be in the lower range. Further, 3–9% of the PBAP tracers reside in the $BSOA_{SRT}$ factor, hence some
of its OC content may rather be attributed to PBAP, further lowering the OC content of the $BSOA_{SRT}$
factor but strengthening coarse OC as a proxy of PBAP. $BSOA_{SRT}$ made a negligible contribution to
fine, coarse and $PM_{10}$ OC in all seasons, except in summer (22–25%), apparently contradicting previous
studies that unambiguously points to BSOA as the major carbonaceous aerosol source at Birkenes in the
vegetative season (Simpson et al., 2007; Yttri et al., 2011b). Note that a prevailing BSOA source in
summer is considered a normal situation also for European rural background environment in general
(e.g. Gelencser et al., 2007), not only for Birkenes. Table 3 shows that $OC_{BSOA,SRT}$ obtained by PMF for
August 2016–2018 is substantially lower than that obtained by LHS for August 2009 (Yttri et al.,
2011b). Although not obtained for the same year, we argue that methodology rather than climatology
explains most of the difference. $OC_{BSOA,LHS}$ provides an upper estimate including all modern carbon,
local and from LRT (excluding biomass burning and PBAP fungal spores and plant debris), whereas
$OC_{BSOA,SRT}$ gives a lower estimate accounting for locally formed BSOA.

It is less likely that anthropogenic secondary organic aerosol (ASOA) resides in the $BSOA_{SRT}$-

factor, as ASOA precursors result from combustion processes and evaporative losses. Further, Yttri et
al. (2011a) found higher ASOA concentrations in the Norwegian rural background environment in
winter compared to summer, which is opposite of $BSOA_{SRT}$, hence co-variation and/or apportionment
to the same factor do not appear likely. ASOA is less abundant than BSOA at Birkenes, as calculated
by Simpson et al. (2007) and Bergström et al. (2012) but the estimates vary substantially and are very
uncertain (Spracklen et al., 2011), particularly for ASOA (from 1% to 10–20%). It is difficult to predict
which PMF factor(s) accounted for ASOA, but for the sake of separating OC into a natural and an
anthropogenic fraction we assume that ASOA is not part of neither $BSOA_{SRT}$ nor the PBAP factor,
which we consider as exclusively natural factors. To provide an upper estimate of the natural sources
(Figure 8), we neither consider it part of the MIN factor.

With 90% (in $PM_{10}$) and 92% (in $PM_{2.5}$) of the MIN factor's OC content attributed to LRT

BSOA ($OC_{BSOA,LRT}$) (See Sect. 2.4.1), the combined contribution of locally formed BSOA ($OC_{BSOA,SRT}$)
and $OC_{BSOA,LRT}$ to OC in $PM_{10}$ and $PM_{2.5}$ would be 34–38% on an annual basis, 37–41% in spring and
50–57% in summer. From this we can deduct that 1/3 of BSOA is of local origin, whereas 2/3 are long-
range transported. For August 2016–2018, the joint contribution of $OC_{BSOA,SRT}$ and $OC_{BSOA,LRT}$ to OC in





$PM_{10}$ is 31%, corresponding better with the LHS estimate (Table 3) but still noticeably lower. Notably,
$OC_{BSOA,SRT}$, $OC_{BSOA,LRT}$ and $OC_{PBAP}$ combined contributed 79% to OC in $PM_{10}$ in August 2016–2018,
which exactly matches the sum of $OC_{BSOA,LHS}$ and $OC_{PBAP,LHS}$ to OC in $PM_{10}$ in August 2009. This
suggests that LHS and PMF apportion an equally large amount of OC to natural sources but that the
split between BSOA and PBAP likely differ. It is evident that the LHS-approach provides an upper
estimate of BSOA (Gelenscer et al., 2007; Yttri et al. 2011a), whereas the great diversity of PBAP likely
is underestimated by just accounting for plant debris and fungal spores. The lower estimate of $OC_{BSOA}$
and the higher estimate of $OC_{PBAP}$ provided by PMF in the present study is in line with this and
encourage further effort to apportion these major carbonaceous aerosol sources correctly. Inclusion of
monoterpene and sesquiterpene oxidation products (Kleindienst et al., 2007) to PMF would possibly
improve our understanding of the SOA apportionment, as would knowledge about their atmospheric
lifetime.

### 4.  Conclusions

The carbonaceous aerosol time-series at the Birkenes Observatory initiated in 2001 is unique due to its
unprecedented length in Europe and because measurements are performed both for $PM_{10}$ and $PM_{2.5}$.
Such long-time series are of utmost importance, e.g. for the evaluation of projections, air-quality models,
and climate models. The need for concurrent and diverse off-line and on-line carbonaceous aerosol
speciation measurements for understanding of carbonaceous aerosol sources, seasonal, annual, and long-
term variability has been utterly demonstrated.
Statistically significant and comparably large reductions ($\sim$ -4% $yr^{-1}$) were calculated for EC
and $PM_{2.5}$ at the Birkenes Observatory for 2001–2018, with EC reductions largely attributed to road
transportation. No significant declining trend was calculated for OC, likely because prevailing natural
sources masked any reduction in anthropogenic sources. Further reduction of carbonaceous aerosol may
be hampered by poorly abated sources such as domestic heating, though more work is needed to assess
this. The OC fraction of $PM_{10}$ (+2.3% $yr^{-1}$) and $PM_{2.5}$ (+3.2% $yr^{-1}$) increased significantly from 2001–
2018, whereas the EC fraction decreased (-4.0 – -4.7% $yr^{-1}$), causing a successive change in the aerosol
chemical composition and in the relative source composition.
Source apportionment using PMF identified seven factors, six of which were carbonaceous
dominated: Mineral dust dominated (MIN), traffic/industrial-like (TRA/IND), biogenic secondary
organic aerosol (BSOA), primary biological aerosol particles (PBAP), biomass burning (BB) and
ammonium nitrate dominated ($NH_4NO_3$). Carbonaceous material was negligible in the sea salt (SS)
factor. Combustion of fossil fuel (78%) was the major source of EC and TRA/IND (50%) the key factor.
Emissions from residential wood burning increased in the heating season but fossil fuel sources
dominated EC even in winter (66%). Continental Europe and western parts of Russia were the main
source regions of elevated levels of eBC, both for biomass burning and for combustion of fossil fuels.
Natural sources dominated both fine (53%) and coarse (78%) fraction OC, thus also OC in $PM_{10}$ (60%).




The natural fraction increased substantially in the vegetative season due to biogenic secondary organic
aerosol and primary biological aerosol particles, confined to the BSOA, PBAP and MIN factors. 77–
91% of OC was attributed to natural sources in summer and 22–37% in winter. The coarse fraction
showed the highest share of natural sources regardless of season and was dominated by PBAP, except
in winter. Notably, PBAP (26%) made a larger contribution to OC in $PM_{10}$ than BB (14%), and an
equally large contribution as BB (17%) in $PM_{2.5}$.

**Author contribution**
Conceptualization, Methodology and Writing – Original draft, Visualization: **Wenche Aas, Stephen M.**
**Platt, Xin Wan, Karl Espen Yttri;** Data Curation: **Markus Fiebig, Hans Gundersen, Anne-Gunn**
**Hjellbrekke, Hilde Uggerud, Marit Vadset, Karl Espen Yttri,;** Formal Analysis: **Stephen M. Platt,**
**Sverre Solberg, Xin Wan, Karl Espen Yttri;** Funding acquisition: **Wenche Aas; Kjetil Tørseth;**
Resources: **Jason Surratt**; Software: **Francesco Canonaco, Sabine Eckhardt, Nikolaos Evangeliou,**
**Stephen M. Platt, André S. H. Prévôt**; Writing, review and editing: **Wenche Aas, Francesco**
**Canonaco, Sabine Eckhardt, Nikolaos Evangeliou, Markus Fiebig, Hans Gundersen, Anne-Gunn**
**Hjellbrekke, Cathrine Lund Myhre, Stephen M. Platt, André S. H. Prévôt, David Simpson, Sverre**
**Solberg, Jason Surratt, Kjetil Tørseth, Hilde Uggerud, Marit Vadset, Xin Wan**, **Karl Espen Yttri**

**Acknowledgements**
Time series used in the present study, except for the organic tracers, were obtained as part of the
Norwegian national monitoring program (Aas et al., 2020). The monosaccharide anhydrides, the sugar-
alcohols and the 2-methyltetrols (organic tracers) time series were funded by the Norwegian Research
Council through the Strategic Institute Projects "*Observation and Modelling Capacities for Northern*
*and Polar Climate and Pollution*" and the "*Studying sources, formation and transport of short-lived*
*climate forcers by advanced high-time resolution measurements*". All data are reported to the EMEP
monitoring programme (Tørseth et al., 2012) and are available from the database infrastructure EBAS
(http://ebas.nilu.no/) hosted at NILU.
The research leading to these results has benefited from Aerosols, Clouds, and Trace gases
Research InfraStructure (ACTRIS), funding from the European Union Seventh Framework Programme
(FP7/2007–2013) under ACTRIS-2 and the grant agreement no. 262254, and the COST Action
CA16109, Chemical On-Line cOmpoSition and Source Apportionment of fine aerosol-COLOSSAL;
I.e., for participation in interlaboratory comparison for thermal-optical analysis and QA/QC of
measurements.
OC/EC and mass concentration were measured as part of the Norwegian national monitoring
programme (Aas et al., 2020), whereas monosaccharide anhydrides were analysed as part of the SACC
(Strategic Aerosol Observation and Modelling Capacities for Northern and Polar Climate and Pollution)
and SLCF (Describing sources, formation, and transport of short lived climate forcers using advanced,



novel measurement techniques) projects.

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





**Figures**

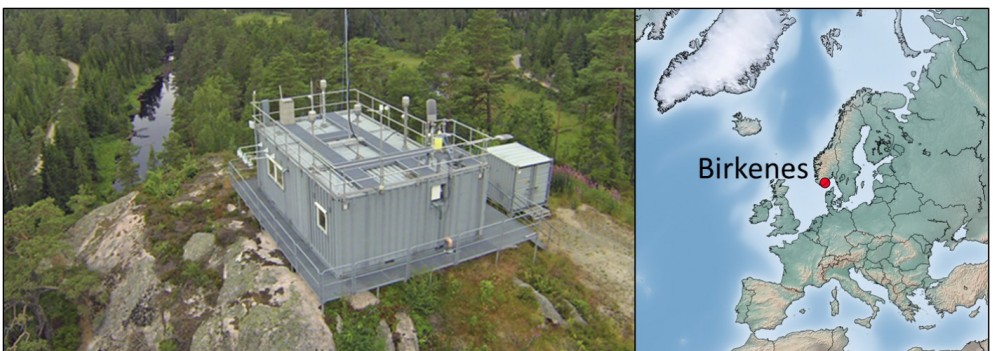


**Figure 1: The Birkenes Observatory (58°23' N, 8°15' E; 219 m asl) lies in the Boreo-nemoral zone, 20 km from the**
**Skagerrak coastline in Southern Norway.**



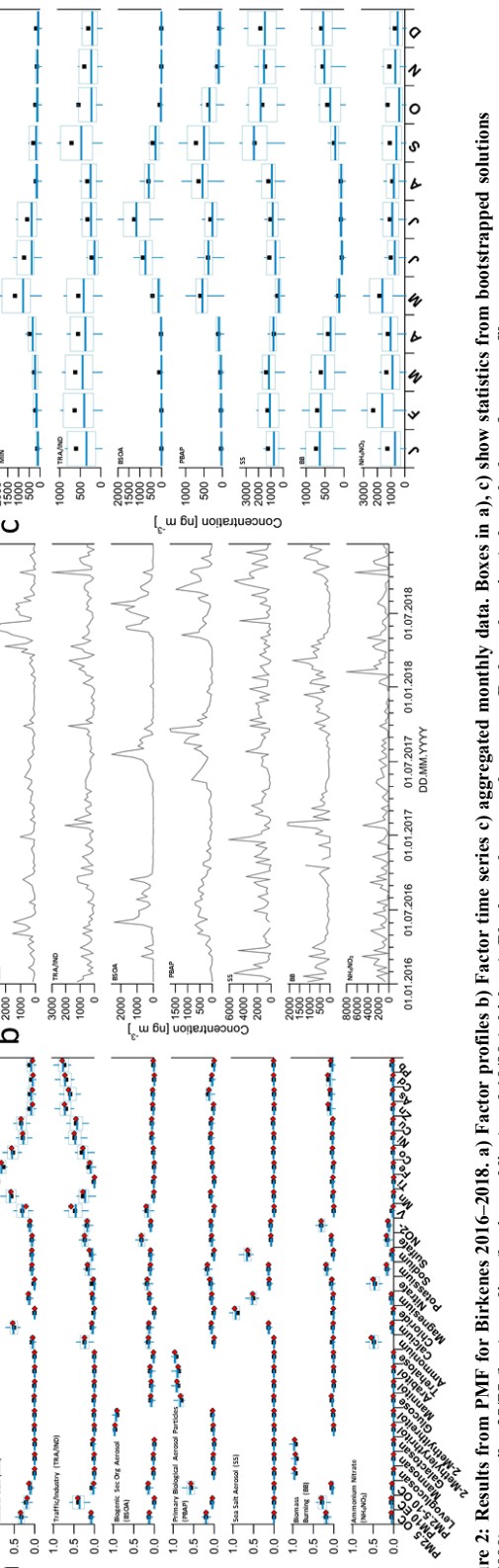

**Figure 2: Results from PMF for Birkenes 2016–2018. a) Factor profiles b) Factor time series c) aggregated monthly data. Boxes in a), c) show statistics from bootstrapped solutions (n=5000): percentiles 25/75 (box), median (horizontal line) and 10/90 (whiskers). Black markers in a) show the means. Red markers in a) show the base factor profiles.**







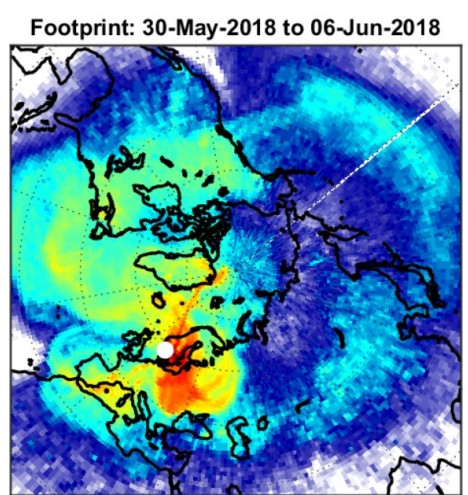


**Figure 3: Footprint emission sensitivities calculated using the FLEXPART model for the period 30 May–6 June 2018 at the Birkenes Observatory.**







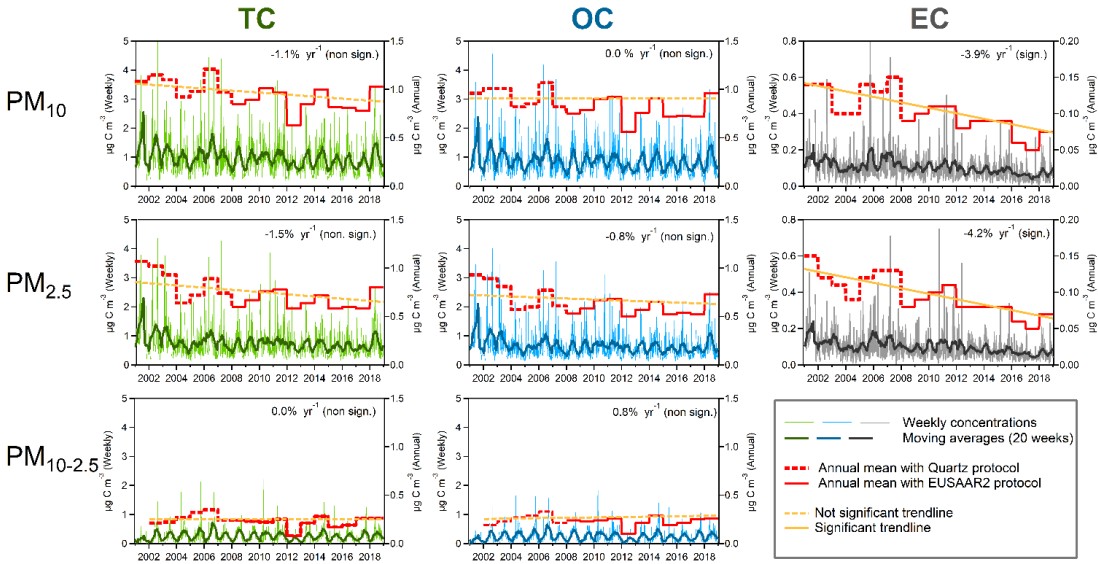


**Figure 4: Ambient aerosol concentrations of TC, OC and EC in PM$_{10}$ (Upper panels), in PM$_{2.5}$ (Mid-Panels), and TC and OC in PM$_{10-2.5}$ (Lower panels), presented as weekly (168 h) and annual mean concentrations for the Birkenes Observatory for 2001–2018. The trendlines account for the protocol shift.**






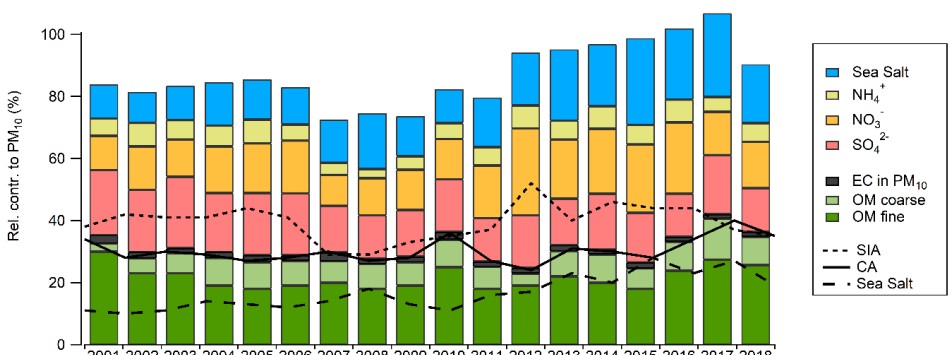


**Figure 5: Mass closure of PM₁₀ for Birkenes for the period 2001–2018 (Unit: %). Notation: Sea salt = Sum of Na⁺, Mg²⁺, Cl⁻; SIA = Secondary inorganic aerosol (SIA) = Sum of SO₄²⁻, NO₃⁻; NH₄⁺); CA = Carbonaceous aerosol; OM = Organic matter. OM is calculated using OC:OM=1.9 (Yttri et al., 2011a).**






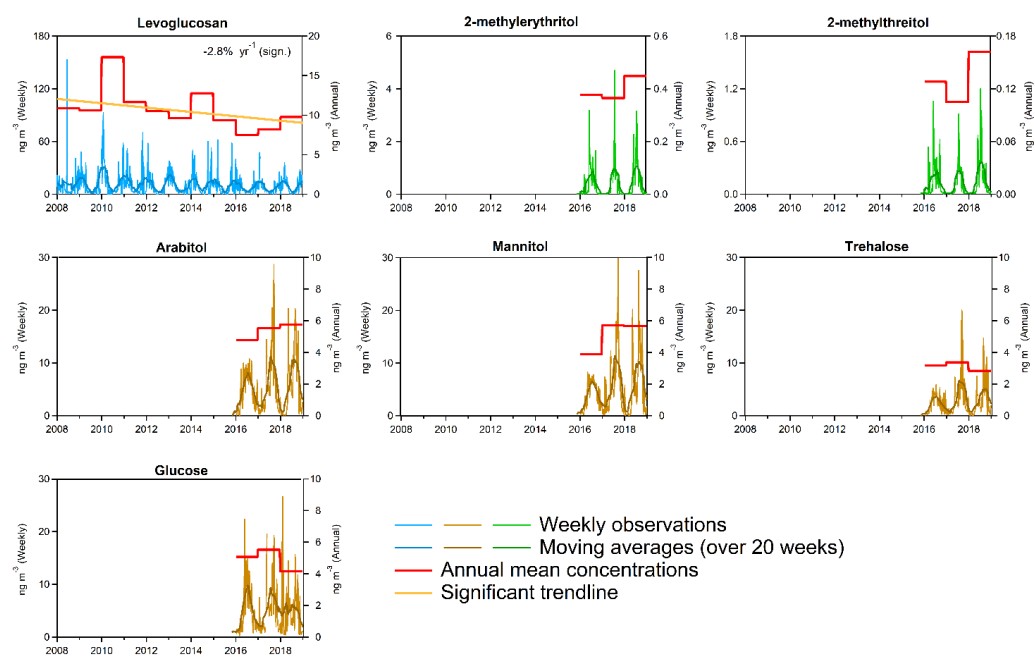


**Figure 6: Ambient aerosol concentrations of organic tracers in PM$_{10}$. Levoglucosan, 2-methylerythritol and 2-methylthreitol (Upper panels), arabitol, mannitol, trehalose (Mid-Panels), and glucose (Lower panel), presented as weekly (168 h) and annual mean concentrations for the Birkenes Observatory for the period 2008–2018.**




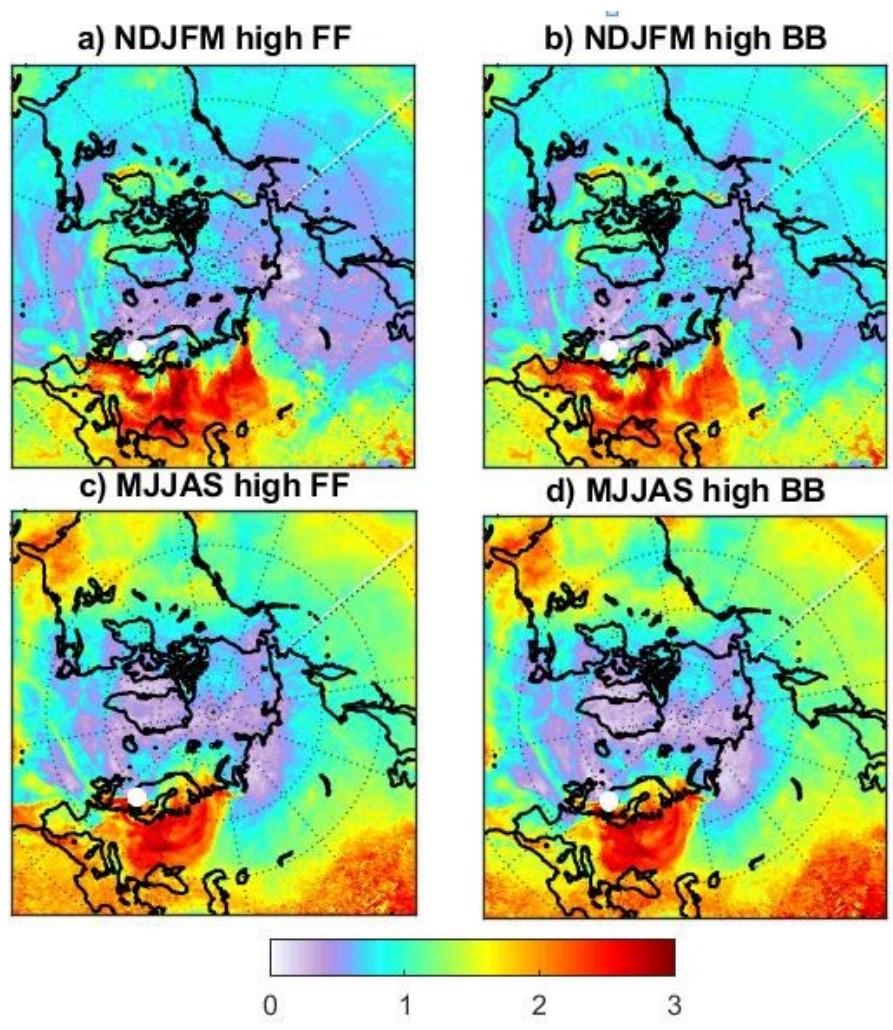

**Figure 7: 70th percentiles of eBC$_{ff}$ (left panels, a and c) and eBC$_{bb}$ (right panels, b and d) for winter (NDJFM) and summer (MJJAS). The color-coding shows the ratio of residence times for footprint sensitivities during measurements exceeding the 70th percentile and the average footprint sensitivity.**



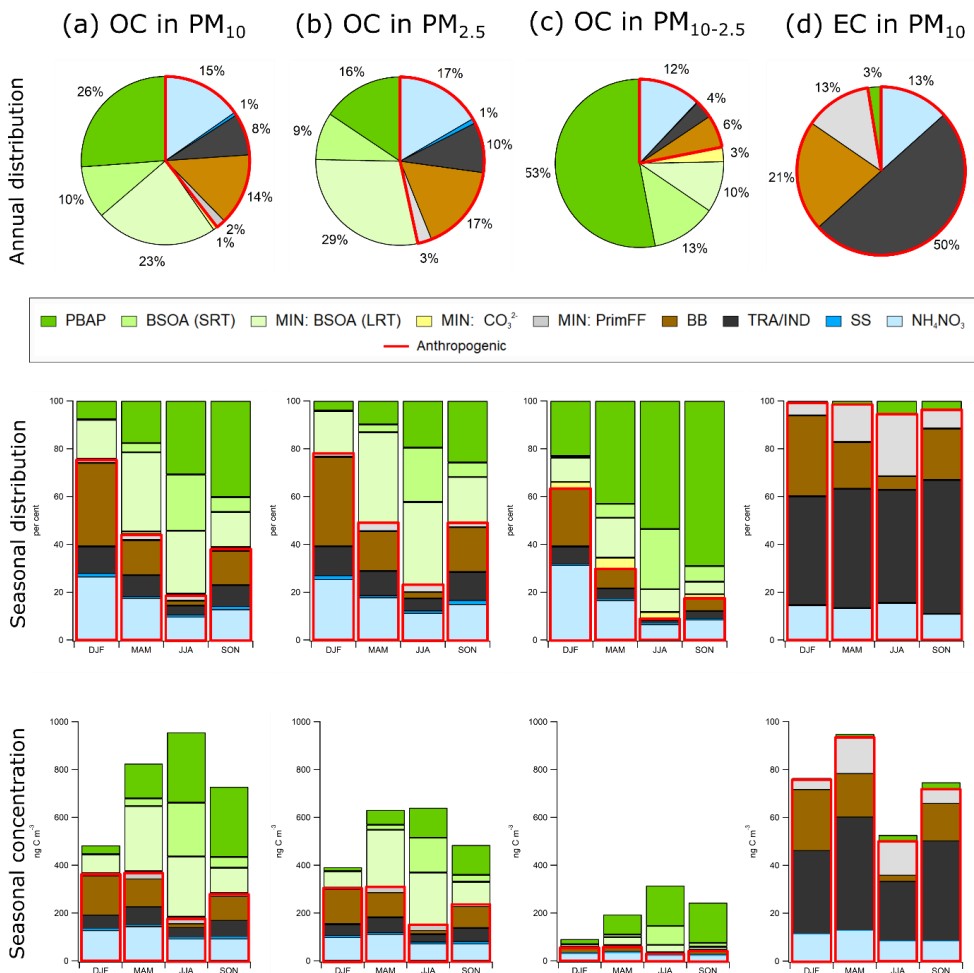

**Figure 8: Factor contributions to OC in PM₁₀ (a), PM₂.₅ (b), PM₁₀₋₂.₅ (c), and EC in PM₁₀ (d) at Birkenes (2016–2018) (upper panels), and divided into seasons (middle and lower panels), as determined by positive matrix factorisation. The factors enclosed by the full red line represents anthropogenic sources. The OC content of the MIN factor is divided into long range transported BSOA (OC_BSOA,LRT) and primary OC from fossil fuel combustion (OC_PrimFF) following Eq. (1), and carbonate carbon (OC_CO32-) (Sect. 2.4.1).**



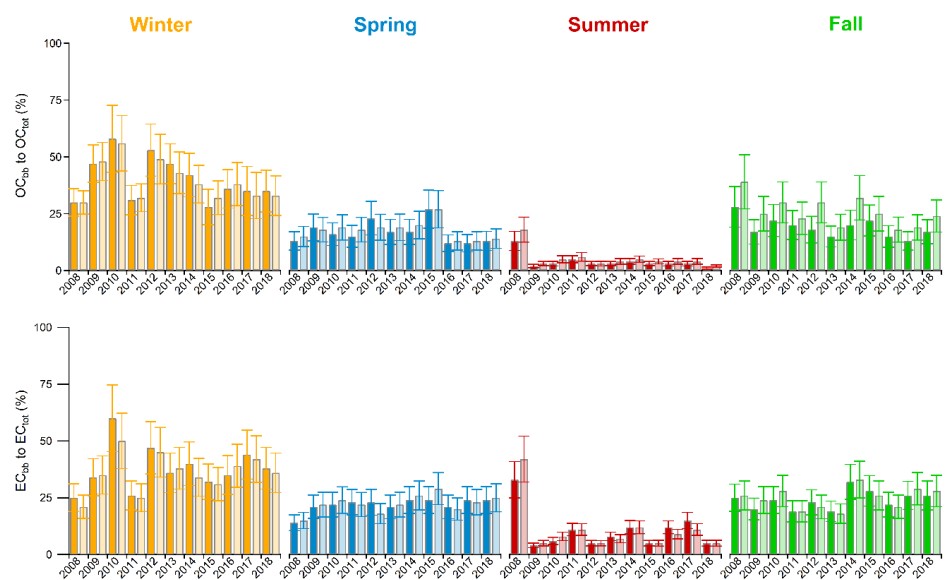

**Figure 9: Relative contribution of OC_bb to OC_tot (upper panel) and EC_bb to EC_tot (lower panel) in PM_10 (dark colors)**
**and in PM_2.5 (light colors), as a function of season at Birkenes for 2008–2018 (DJF = Winter; MAM = Spring; JJA =**
**Summer; SON = Fall).**







**Tables**
**Table 1: Variables describing the biomass burning source derived from the PMF BB factor in the present study, and**
**comparable variables obtained by [14]C-analysis reported by Zotter et al. (2014).**

| Present study | Zotter et al. (2014)[1] |
|---|---|
| OC/Levoglucosan (in $PM_{10}$) = 12.7 | $OC_{NF}$/Levoglucosan (in $PM_{10}$) = 12.6 ± 3.1 |
| OC/Levoglucosan (in $PM_{2.5}$) = 11.1 | |
| EC/Levoglucosan (in $PM_{10}$) = 1.96 | $EC_{NF}$/Levoglucosan (in $PM_{10}$) = 1.72 ± 0.59 |
| OC/EC (in $PM_{10}$) = 6.5 | $OC_{NF}/EC_{NF}$ (in $PM_{10}$) = 7.7 ± 2.1 |
| OC/EC (in $PM_{2.5}$) = 5.7 | |

[1]North of the Alps
Notation: $OC_{NF}$ = Non-fossil OC; $EC_{NF}$ = Non-fossil EC






**Table 2: Biomass burning fraction derived from the PMF and the aethalometer model. Aethalometer model 1 shows**
**the biomass burning fraction obtained by the default pair of Absorption Ångstrøm Exponents (AAE) suggested by**
**Zotter et al. (2014), whereas Aethalometer model 2 show the biomass burning fraction obtained using the pair of AAEs**
**derived from PMF.**

|  | **PMF** | **Aethalometer model 1** | **Aethalometer model 2** |
|---|---|---|---|
| Biomass burning fraction | 0.27 | 0.48 | 0.28 |
| Fossil AAE | 0.93 | 0.9 (Zotter et al. 2017) | 0.93 (from PMF) |
| Biomass burning AAE | 2.04 | 1.68 (Zotter et al. 2017) | 2.04 (from PMF) |





**Table 3:** $OC_{BSOA}$, short-range transported (SRT) and long-range transported (LRT), and $OC_{PBAP}$ concentrations and their relative contribution to OC in $PM_{10}$ at Birkenes in August, as obtained by Latin Hypercube Sampling (Yttri et al., 2011b) and by PMF (present study).

| | Reference | Approach | $OC_{BSOA,SRT}$ (ng C m⁻³) | $OC_{BSOA,SRT}$/OC | $OC_{BSOA,SRT+LRT}$ (ng C m⁻³)[2,3] | $OC_{BSOA,SRT+LRT}$/OC[2,3] | $OC_{PBAP}$ (ng C m⁻³)[2] | $OC_{PBAP}$/OC[2] |
|---|---|---|---|---|---|---|---|---|
| August 2009 | Yttri et al. (2011b) | LHS[3] | 115 | | 505[1] (408–598)[2,3] | 0.48[1] (0.38–0.58)[2,3] | 290[1] (213–380)[2] | 0.31[1] (0.22–0.40)[2] |
| August 2016 | Present study | PMF | 183 | 0.19 | 173 | 0.28 | 318 | 0.52 |
| August 2017 | Present study | PMF | 159 | 0.19 | 252 | 0.26 | 553 | 0.57 |
| August 2018 | Present study | PMF | 159 | 0.20 | 316 | 0.40 | 287 | 0.36 |
| August 2016–2018 | Present study | PMF | 152 | 0.19 | 247 | 0.31 | 386 | 0.48 |

1. 50th percentile
2. 10th–90th percentile
3. LHS-approach includes both $OC_{BSOA,SRT}$ and $OC_{BSOA,LRT}$
