# Peer review of "Trends, composition, and sources of carbonaceous aerosol at"

_Atmospheric Chemistry and Physics, 2020_

## Referee Comment (RC1) · Anonymous Referee #1 · 18 Jan 2021

This manuscript presents and discusses results from a valuable long-term study of carbonaceous aerosol for the Birkenes Observatory in Southern Norway. A thorough data analysis was performed and the results are of substantial interest to the aerosol community. Unfortunately, as indicated below, the manuscript suffers from several shortcomings. Major revision is certainly needed before it can be published in ACP.

Major comments:

1. Previous long-term aerosol studies at the study site deserve to be referenced to. I am thinking of Amundsen et al., Atmos. Environ. A, 26 (1992) 1209-1324 and especially of Maenhaut, Nucl. Instrum. Methods B, 417 (2018) 133-138. In the latter

study, the PM mass, BC (which can serve as a proxy for EC) and 21 elements were determined in PM2 samples over a 5-year period (1991-1996) and the results were subjected to PMF analysis, using EPA PM5, and 8 factors were extracted, with one of them being wood burning, which accounted for 14% of the PM2 mass. It would be of interest to see a comparison of the PM mass and PMF data from that study with the corresponding data for the PM2.5 aerosol fraction of the current study. For one thing, biomass burning (BB) accounts for 17% of the PM2.5 OC (Figure 8); from Table S 7, one can derive that the OC/PM2.5 ratio is around 0.2, so that BB accounts for only 3.5% of the PM2.5 mass; this percentage is very much lower than the 14% in the study of Maenhaut (2018). Is there any sensible explanation for this large discrepancy?

2. Abbreviations and acronyms should be defined (written full-out) when first used in the Main text or the Supplement, and they should only be defined once. This applies to the following:

Line 79: BSOA; it is only defined in lines 659-660;

Line 187: PMF; it is already defined in line 124;

Line 228: BB; it is only defined in lines 419 and 660;

Line 303: SS; it is only defined in lines 320, 419-420, and 661;

Line 319: LRT; it is already defined in lines 217-218;

Line 395: CI;

Supplement, line 44: LOD.

3. The manuscript is on some occasions unclear and/or confusing.

In line 134 the authors mention the coordinates of the Birkenes Observatory, but then in line 175 they talk about the old and new Birkenes sites. This is confusing; both sites should be mentioned in line 134, it should be indicated what the distance between both sites was and in which year the measurements in the new site were started. According

to a NILU Website, the observatory was moved to a new building in 2009.

It should be clearly indicated in sections 2.4 and S4 that the PMF on the aerosol data was carried out on the data set of 2016-2018 and I suggest that the number of samples (apparently 151, Table S 3) is indicated in both sections. Furthermore, a literature reference for the PMF approach is needed; I presume that use was made of EPA PMF5.

It seems that the PM mass data were not included in the PMF analysis. Why not? Including them would allow to assess the contribution of the different factors to it, which is very useful information.

It is unclear what the difference is between the base factor profiles and the black square markers in Figure 2a.

The legend of Figure S 3 should be extended. I suggest inserting "on the filter data" between "solution" and "presented". Also the headings of Tables S 3 and S 4 should be extended; it should be indicated that they are for the filter data. I also suggest to indicate in the legend and in the two captions that the filter data set of 2016-2018 was used.

4. The manuscript needs to better organized. Section 2.4.1 does not belong in 2. Methodology, it should be moved to 3. Results and discussion. Also, is it not possible to combine this section and section 3.2 into one single section?

5. The manuscript has several grammatical errors; often the subject is in plural and the verb in singular (or vice versa).

6. There are major problems with the references, both in the Main text and in the Supplement.

The initials of the authors should be consistently after the authors' last names.

Titles of journal articles should be in lower case, not in Title Case.

Abbreviated journal names are needed throughout.

The publication year should be at the end of each reference and it should not be in parentheses.

For the Main Text:

"Denier van der Gon" should be replaced by "van der Gon" both in the text and in the Reference list, and the reference should be moved down in the reference list.

Line 707: First names of the authors should be replaced by initials.

Lines 959-968: "Myhre and Samset, 2015" should come before "Myhre et al., 2013"

Lines 1067-1070 should be deleted.

The following references of the text are missing in the Reference list:

Line 67: Pio et al., 2007;

Line 108: Sillanpää et al., 2006; there is "Sillanpää et al., 2005" in the Reference list, but there is not referred to this within the text.

Line 109: Genberg et al., 2013;

Line 176: Mann, 1945;

Line 177: Kendall, 1975;

Lines 177 and 178: Gilbert, 1987;

Line 178: Theil, 1958; perhaps this should be "1950" instead of "1958, see below;

Line 178: Sen, 1968;

Line 215: Polissar et al., 1998;

Line 215: Norris et al., 2014;

Line 241: Pacyna et al., 1996;

Line 305: Stumm and Mogan, 1995; there is "Stumm and Morgan, 1996" in the Reference list, but there is not referred to this within the text.

Line 620: Spracklen et al., 2011.

There is no reference made in the Main text to the following references that are in the Reference list:

Aas et al., 2019;

Birch and Cary, 1996;

Cattiaux et al, 2010;

Fine et al, 2001; 2002a, 2002b, 2004;

Hu et al, 2018;

Jordan and Seen, 2005;

Long et al., 2013;

Putaud et al, 2010;

Schmidl et al, 2008;

Theobald et al. 2019;

Turpin and Lim; 2001;

Tørseth et al, 2000;

Winiwarter et al. 1999.

For the Supplement:

References that are mentioned in its text should also be included in the Reference list,

even if they are already in the Reference list of the Main text.

Line 230: The title of the journal article should be included.

Lines 233 and 254: Is it Backman or Backmann? In lines 116 and 118 there is Backmann, but in line 154 Backman.

The following references of the text are missing in the Reference list:

Line 7: Yttri et al., 2007a;

Line 10: Yttri et al., 2011; also, this should either be 2011a or 2011b or even 2011c;

Lines 10-11: Yttri et al., 2019;

Line 18: Cavalli et al., 2010;

Line 23: Subramanian et al., 2006; there is "Subramanian et al., 2004" in the Reference list, but there is not referred to this within the text

Line 57: Theil, 1958; There is "Theil, 1950" in the Reference list, and it looks that this is the correct year.

Line 60: Tørseth et al, 2012;

Line 61: Aas et al. 2019;

Line 125: Sandradewi et al., 2008;

Line 137: Paatero et al., 1994; there is "Paatero and Tapper, 1994" in the Reference list, but there is not referred to this within the text.

Line 206: Zotter et al., 2014;

Line 221: Bauer et al., 2008a (incidentally, the "a" should be removed);

Line 221: Yttri et al., 2007a;

Lines 221-222: Yttri et al., 2011a,b.

[Figure]

There is no reference made in the text to the following reference that is in the Reference list:

Pio et al., 2007.

Minor comments and corrections for the Main Text:

Line 21: Replace "organic-" by "organic".

Line 23: Replace "-at the" by "at the".

Lines 27-30: It says "six" in line 27, but then in the remainder of the sentence, 7 components are listed. This is confusing. In lines 415-417 it is stated that only 6 out of the 7 components were associated with significant amounts of carbonaceous aerosol.

Line 72: Replace "and fungus" by "and fungi".

Line 90: Replace "biogenic," by "biogenic;".

Line 118: Replace "have developed" by "has developed".

Line 170: Replace "monosaccharides anhydrides" by "monosaccharide anhydrides".

Line 182: Replace "Scientific" by "Scientific)".

Line 184: Replace "in which" by "which".

Line 209: Replace "principal" by "principle".

Line 217: Replace "is via" by "are via".

Line 292: Replace "also favours" by "also favour".

Line 305: Replace "resembles these" by "resemble these".

Line 334: Replace "EC, (Sect. 3.1.2) but" by "EC (Sect. 3.1.2), but".

Line 349: Replace "Table S 8" by "Table S 8)".

Line 363: The reference to Table S 4 here is unclear; I presume that reference should be made to another Table.

Lines 395-396: The data listed here apparently relate to PM2.5 and PM10; it should be specified which ones are for which size fraction. Also in line 396, replace "-1.8, CI" by "-1.8%, CI".

Line 455: Replace "et al.," by "et al.".

Line 472: Replace "airmasses" by "air masses".

Line 530: Replace "2003)" by "2003".

Line 605: Replace "points to" by "point to".

Line 619: Replace "(2012) but" by "(2012), but".

Line 634: Replace "likely differ" by "likely differs".

Line 698: Replace "I.e." by "i.e.".

Line 714: Replace "Sci Rep" by "Sci. Rep.".

Line 786: Replace "organicaerosols" by "organic aerosols".

Line 904: Replace "5," by "5, 5065,".

Line 970: Replace "O'dowd" by "O'Dowd".

Line 1019: Replace "Sioutas" by "and Sioutas".

Line 1106: Replace "B.J., Lim" by "B. J. and Lim" in case this reference is retained.

Line 1120: Replace "www.atmos-chem-phys.net/12/5447/2012/doi:10.5194/acp-12-5447-2012," by "https://doi.org/10.5194/ acp-12-5447-2012,".

Line 1139: Replace "pri-mary" by "primary".

Line 1187: Replace "Atmos. Chem. Phys. Discuss., 14, pp. 15591-15643," by "Atmos.

Chem. Phys., 14, 13551-13570, https://doi.org/10.5194/acp-14-13551-2014,".

Lines 1192-1193: Replace "www.atmos-chem-phys.net/17/4229/2017/doi:10.5194/acp-17-4229-2017," by "https://doi.org/10.5194/ acp-17-4229-2017,".

Line 1215: Replace "); CA" by "; CA".

Line 1251, within Table 2: Replace twice "Zotter et al. 2017" by "Zotter et al., 2017".

Minor corrections for the Supplement:

Line 41: Replace "is text not" by "text are not".

Line 64: Replace "often deviates" by "often deviate".

Line 83: Replace "was applied" by "were applied".

Line 116: Replace "by of" by "by".

Line 117: Replace "adapt the" by "adopt the".

Line 194: Replace "then then" by "then".

Line 214: Replace ") , and" by "), and".

g)    ") , and" was replaced by "), and". (We found this in line 195, not 214)

Line 240: Replace "2013" by "2013.".

Line 343: Replace "1389.https" by "1389, https".

Line 394: Replace "bar (Figure S 2)" by "bar".

Line 396: Replace "were only" by "was only".

Line 404: Replace "for their identification" by "for identification".

l)    "for their identification" was replaced by "for identification".

Line 404, within Table S 1: Replace "Methylterythritol" by "Methylerythritol".

---

## Referee Comment (RC2) · Anonymous Referee #2 · 18 Jan 2021

The present study investigates the carbonaceous composition of PM2.5 and PM10 obtained at the Birkenes Observatory (GAW - EMEP) from 2001 to 2018. The data series is unique in Europe and has invaluable scientific interest. The treatment and interpretation of the data is adequate and the results and conclusions obtained are very relevant.

The results demonstrated a long-term change in the chemical composition of the aerosol at the background site of Birkenes and therefore also in the sources contribution to PM.

Authors applied PMF receptor model to the 2016-2018 chemical dataset identifying

6 carbonaceous aerosol sources. They demonstrated a decrease of traffic and industry OC/EC emissions, while the abatement of OC/EC from biomass burning has been slightly less successful. Moreover, results emphasize the importance of biogenic sources (BSOA and PBAP) at Birkenes site. The results demonstrated a decrease in OC / EC emissions from traffic and industry affecting the Birkeness site. This decrease is not as obvious for OC / EC from biomass burning, indicating a less successful abatement strategy. The results also emphasize the importance of biogenic sources (BSOA and PBAP) at the Birkenes site.

Authors concluded the need of investigating trends in levoglucosan and biomass burning in Europe, given the further importance residential wood burning as a major source of air pollution in Europe. The need of measuring monoterpene and sesquiterpene oxidation products for improving the SOA apportionment is also highlighted. These data can be essential to improve the model outputs.

Minor correction. The data regarding the decreasing EC fractions in PM2.5 (-4.0% yr-34 1) and PM10 (-4.7% yr-1) in the abstract differs from the data in the main text (Line 380: whereas it decreased for the EC fraction (-3.9 – -4.5% yr-1)).

Line 58. I think OC and EC are not regularly measured in Air Qualtiy Monitoring networks, with the exception of EMEP/GAW and aerosol in.situ ACTRIS sites.

Method section

2.2.1. What is the difference between the old and the new Birkenes Observatory? When was the new observatory installed? Please add some comments on 2.1.

2,2,2 - Lines 187 188- It should be noted here that the "novel" positive matrix factorization for BC data is based on the methodology devised by Platt et al, in preparation. Is there any other reference to this method available (eg conference proceedings....)?

2.4. It should be noted here that the PMF was applied to the 2016-2018 dataset. Line 210: was OC in PM2.5-10 calculated by difference (OCPM10 – OC PM2.5)? Was EC

inPM25 equal (or similar) to EC in PM10?

---

## Short Comment (SC1) · 6 Feb 2021

The work by Yttri et al. presents a great, valuable summary of 18 years of organic aerosol measurements.

The absent organic carbon (OC) trend is most interesting to me. I believe this could be highlighted even more in abstract and conclusions. What do other OC trends show in Europe? Is this the only work attempting to calculate OC trends at European sites? Could the present work be put a bit more in a European context? More references?

A side question: What influence has the change of the OC measurement protocol in

the middle of the period on the OC trend? In what direction can this have influenced the trend?

---

## Author Comment (AC1) · 18 Mar 2021

The authors own Action items:

We have made the following changes and additions to the paper separate from the comments made by Referee #1 and #2 and the short comment made by Martin Schulz:

We realized that the title of our revised manuscript could be improved, making the following change: "Trends, composition, and sources of carbonaceous aerosol at the Birkenes Observatory, Northern Europe, 2001-2018. Hence, it would be possible for a future reader to know the time span the paper covers by just reading the title. For sure,

the phrase "last 18 years", used in the original title is imprecise. (Line 1-2) and (Line 1-2 in Suppl.)

"TRA/IND" was stated the other way around (IND/TRA) three places in the manuscript, we have changed this so that all reads "TRA/IND".

We have switched from "Positive Matrix Factorisation" to "Positive Matrix Factorization" due to the US origin of this analytical approach and for consistency throughout the manuscript.

---

## Author Comment (AC2) · 18 Mar 2021

Reply to referee #1: Major Comment by R1: 1. Previous long-term aerosol studies at the study site deserve to be referenced to. I am thinking of Amundsen et al., Atmos. Environ. A, 26 (1992) 1209-1324 and especially of Maenhaut, Nucl. Instrum. Methods B, 417 (2018) 133-138. In the latter study, the PM mass, BC (which can serve as a proxy for EC) and 21 elements were determined in PM2 samples over a 5-year period (1991-1996) and the results were subjected to PMF analysis, using EPA PM5, and 8 factors were extracted, with one of them being wood burning, which accounted for 14% of the PM2 mass. It would be of interest to see a comparison of the PM mass and PMF

data from that study with the corresponding data for the PM2.5 aerosol fraction of the current study. For one thing, biomass burning (BB) accounts for 17% of the PM2.5 OC (Figure 8); from Table S 7, one can derive that the OC/PM2.5 ratio is around 0.2, so that BB accounts for only 3.5% of the PM2.5 mass; this percentage is very much lower than the 14% in the study of Maenhaut (2018). Is there any sensible explanation for this large discrepancy?

Reply to Major Comment 1: First, we would like to thank referee 1 (R1) for his/her thorough work going through our manuscript, it is much appreciated!

a) R1 would like to see the following two references included to the manuscript: Amundsen et al., Atmos. Environ. A, 26 (1992) 1209-1324 and Maenhaut, Nucl. Instrum. Methods B, 417 (2018) 133-138.

Action: We have included the following sentence in line xxx-xxx and Aamundsen et al. (1992) and Maenhaut (2018) have been added to the reference list: Two previous studies have used factor analysis to study PM sources at Birkenes (Aamundsen et al. (1992; Maenhaut, 2018). The present study focuses on carbonaceous aerosol, using OC, EC and highly source specific organic tracers as input in addition to inorganic species and elements used by Aamundsen et al. (1992) and Maenhaut (2018). This provides a different set of factors, based on different input, hampering any reliable comparison of these studies.

b) R1 states that: "It would be of interest to see a comparison of the PM mass and PMF data from that study (here: Maenhaut, 2018) with the corresponding data for the PM2.5 aerosol fraction of the current study." Yttri et al. (current study) conducted a PMF based source apportionment study of the carbonaceous aerosol (here: OC and EC) and not the PM mass concentration, i.e., the mass concentration is not included in the PMF analysis by Yttri et al. This was a deliberate choice made to support our interpretation of the long time series of OC and EC at the Birkenes Observatory; thus, our focus is on the carbonaceous aerosol and not the PM mass concentration. Even

if we at some time in the future were to redo the PMF analysis, using the same data as in Yttri et al., and by adding the PM mass concentration, a comparison of that study and that of Maenhaut (2018), would still be challenging. This is particularly because the chemical species used as input is so different, resolving different factors both with respect to number and content, thus severely questioning to what extent they can be compared. Both Maenhaut (2018) and Yttri et al. claim to resolve a biomass burning factor, a sea salt factor, and a crustal factor (other factors differ), but their chemical content is still quite different. Using the biomass burning factor as an example: The BB profile in Maenhaut (2018) explains a substantial amount of As (>40%) and >20% Pb, whereas this is not the case for Yttri et al. One possible way to interpret this difference is that the biomass burning factor by Maenhaut (2018) also includes emissions from coal burning. If so, it would it be questionable if these two factors could be compared although they are named the same.

c) R1 states that: (. . .) biomass burning (BB) accounts for 17% of the PM2.5 OC (Figure 8); from Table S 7, one can derive that the OC/PM2.5 ratio is around 0.2, so that BB accounts for only 3.5% of the PM2.5 mass; this percentage is very much lower than the 14% in the study of Maenhaut (2018). Is there any sensible explanation for this large discrepancy?

As stated in b): Can we be sure that we compare the same thing although they have the same name? Are the factors to an equally large degree separated from other factors to allow for this? R1 asks specifically for a comparison of the relative contribution of BB to PM2.5 obtained in the present study (Yttri et al.) and that of Maenhaut (2018), stating that only 3.5% of the PM2.5 in Yttri et al. is attributed to PM from biomass burning whereas the biomass burning factor accounts for 14% of PM2.5 in Maenhaut (2018), and if there any sensible explanation for this large discrepancy? To make a comparison with Maenhaut (2018) with respect to the BB to PM2.5 ratio, asked for by RC1, we must convert OCBB and ECBB (Unit: $\mu$g C m-3) obtained in the present study to BB mass concentration (Unit: $\mu$g m-3). This can be done in several ways, but we

use an approach as similar as that outlined by R1 as possible.

OCBB to OC in PM2.5 (2016-2018): 17% (Figure 8)

ECBB to EC in PM10 (2016-2018): 21% (Figure 8) (ECPM10 Ì ̌ ECPM2.5)

Mean OC concentration in the PM2.5 size fraction (2016-2018): 0.596 $\mu$g C m-3 (Table S5)

Mean EC concentration in the PM2.5 size fraction (2016-2018): 0.063 $\mu$g C m-3 (Table S5)

Calculated OCBB concentration in the PM2.5 size fraction: 0.17 $\times$ 0.596 $\mu$g C m-3 = 0.101 $\mu$g C m-3.

Calculated ECBB concentration in the PM2.5 size fraction: 0.21 $\times$ 0.063 $\mu$g C m-3 = 0.013 $\mu$g C m-3.

Conversion factor OCBB:OMBB, e.g., 2.2 – 2.6, (Turpin and Lim, 1994).

Conversion factor for ECBB to account for other elements then C: 1.1 (Kiss et al., 2002).

Calculated OMBB concentration in the PM2.5 size fraction: 0.101 $\mu$g C m-3 $\times$ 2.2 – 2.6 = 0.222 – 0.263 $\mu$g m-3.

Calculated ECBB concentration in the PM2.5 size fraction: 0.013 $\mu$g C m-3 $\times$ 1.1 = 0.015 $\mu$g m-3.

PM2.5 concentration (2016-2018): 2.5 $\mu$g m-3 (Table S10)

Relative contribution of OMBB and ECBB (in PM2.5) to PM2.5 mass concentration: (0.222 – 0.263 $\mu$g m-3) + 0.015 $\mu$g m-3 $\times$ 100/2.5 $\mu$g m-3 = 9.5 – 11.1%

Based on the numbers presented in Yttri et al., 9.5 – 11.1% of PM2.5 mass concentration can be attributed to the biomass burning source. Although biomass burning emission are dominated by the carbonaceous fraction, there is also an inorganic frac-

tion of unknown quantity that is not accounted for in the calculations above, which will increase the BB fraction of PM2.5 somewhat. Based on these calculations we conclude that the 9.5 – 11.1% BB contribution is not very different from the 14% estimated by Maenhaut (2018). The 3.4% calculation made by RC1 does not seem to account for the fact that OC must be converted to OM to make a comparison with the BB factor presented by Maenhaut (2018), nor that EC must be included in the calculation. If we follow the same outline as RC1 and take these considerations into account, we get the same range as calculated above, i.e., 9.5 – 11.1%.

Major Comment by R1: 2. Abbreviations and acronyms should be defined (written full-out) when first used in the Main text or the Supplement, and they should only be defined once. This applies to the following: Line 79: BSOA; it is only defined in lines 659-660 (Added) Line 187: PMF; it is already defined in line 124 (Removed) Line 228: BB; it is only defined in lines 419 and 660 (Added) Line 303: SS; it is only defined in lines 320, 419-420, and 661 (added) Line 319: LRT; it is already defined in lines 217-218 (removed) Line 395: CI; Supplement, line 44: LOD. (explained)

Reply to Major Comment 2:

Action Major Comment by R1:

3. The manuscript is on some occasions unclear and/or confusing. In line 134 the authors mention the coordinates of the Birkenes Observatory, but then in line 175 they talk about the old and new Birkenes sites. This is confusing; both sites should be mentioned in line 134, it should be indicated what the distance between both sites was and in which year the measurements in the new site were started. According to a NILU Website, the observatory was moved to a new building in 2009.It should be clearly indicated in sections 2.4 and S4 that the PMF on the aerosol data was carried out on the data set of 2016-2018 and I suggest that the number of samples (apparently 151, Table S 3) is indicated in both sections. Furthermore, a literature reference for the PMF approach is needed; I presume that use was made of EPAPMF5. It seems that

the PM mass data were not included in the PMF analysis. Why not? Including them would allow to assess the contribution of the different factors to it, which is very useful information. It is unclear what the difference is between the base factor profiles and the black square markers in Figure 2a..The legend of Figure S 3 should be extended. I suggest inserting "on the filter data "between "solution" and "presented". Also the headings of Tables S 3 and S 4 should be extended; it should be indicated that they are for the filter data. I also suggest to indicate in the legend and in the two captions that the filter data set of 2016-2018 was used

Reply to Major Comment 3: a) R1 states that: In line 134 the authors mention the coordinates of the Birkenes Observatory, but then in line 175 they talk about the old and new Birkenes sites. This is confusing; both sites should be mentioned in line 134, it should be indicated what the distance between both sites was and in which year the measurements in the new site were started.

According to the request made by R1, we have mentioned the new Birkenes Observatory and the old Birkenes site in one line. Distance between the old site and the new Observatory has been stated, as has the year that the measurements at the new Birkenes Observatory was initiated. Action: ".... situated 100 m south-east of the old Birkenes site, initiating measurements in 2009."

b) R1 states that: It should be clearly indicated in sections 2.4 and S4 that the PMF on the aerosol data was carried out on the data set of 2016-2018 and I suggest that the number of samples (apparently 151, Table S 3) is indicated in both sections. Furthermore, a literature reference for the PMF approach is needed; I presume that use was made of EPAPMF5. It seems that the PM mass data were not included in the PMF analysis. Why not? Including them would allow to assess the contribution of the different factors to it, which is very useful information.

According to the request made by R1, we have included the period (2016-2018) for which the PMF analysis was performed both in section 2.4 and S4, as well as added

the number of samples subjected to the PMF analysis. We have also added a reference (ref) to the PMF-approach used. We agree with R1 that including PM mass concentrations to the PMF analysis would allow for apportioning the PM mass concentration to the various factors resolved and that this is useful information. As stated in reply b) to Major Comment 1, we made a deliberate choice in not including the mass concentration to the PMF analysis, as Yttri et al. is a source apportionment study dedicated to the carbonaceous aerosol and not PM mass. We are aware that quite a few PMF studies include PM mass concentration in their analysis, but in the present study we also present a nearly two decades long time series of OC in PM10, PM2.5 and PM10-2.5 and EC in PM10 and PM2.5, a one decade time series of levoglucosan in PM10, trend studies of these, as well as a novel PMF-approach to apportion eBC into a biomass and fossil fraction. Consequently, there is a limit to how much can be included in a paper and still allow for a thorough presentation and discussion of the data. In our opinion, a PMF analysis including the PM mass concentration would generate enough material for separate paper, e.g., Maenhaut (2018).

Action: We performed PMF ME2 (Canonaco et al., 2013) (See Sect. S3 for a description of the analysis principal and S4 for its application to filter data) for samples collected in 2016-2018 (151 samples),

Major Comment by R1: 4. The manuscript needs to better organized. Section 2.4.1 does not belong in 2.Methodology, it should be moved to 3. Results and discussion. Also, is it not possible to combine this section and section 3.2 into one single section?

Reply to Major Comment 4: a) Section 2.4.1 was originally placed before section 3.2. as stated by R1, but was moved to 2.4.1 because several co-authors found it more convenient that the various factors were defined before we start discussing them (It can also be argued whether defining a PMF factor is really a discussion topic). This way we go from discussing the OC and EC time series and their trends in 3.1 to discussing their sources in 3.2 without having to go through a rather technical chapter (2.4.1) on what defines the different factors. We think this improve the readability of the discussion

(and the paper in general) in quite a favorable way. Thus, we would like to argue that the current sectioning of the paper remains the way it is at present.

Major Comment by R1: 5. The manuscript has several grammatical errors; often the subject is in plural and the verb in singular (or vice versa).

Reply to Major Comment 5: a) A native Englishman has read through the paper looking for such incidences. The following corrections were made:

Action: Show changes that was made to the text to account for this.

Major Comment by R1: 6. There are major problems with the references, both in the Main text and in the Supplement. The initials of the authors should be consistently after the authors' last names. Titles of journal articles should be in lower case, not in Title Case. Abbreviated journal names are needed throughout. The publication year should be at the end of each reference and it should not be in parentheses.

Reply to Major Comment 6: a) Indeed, there are (major problems)! We do not know what has caused this and we sincerely apologize. The referee should certainly use most of his/her time when reviewing a manuscript on the scientific content and not typos and poor referencing. We have collected and formatted the references in EndNote for this revised version of the manuscript, which should substantially improve the quality. Further, we have picked the journal abbreviations from https://images.webofknowledge.com/images/help/WOS/J_abrvjt.html. We nevertheless thank R1 for taking on this substantial work when reviewing the manuscript, it is much appreciated!

"Denier van der Gon" should be replaced by "van der Gon" both in the text and in the Reference list, and the reference should be moved down in the reference list.

b) One of the co-authors have been in contact with Dr. Denier van der Gon and has been informed that "Denier van der Gon" indeed is the surname and not "van der Gon". Thus, we keep Denier van der Gon.

Line 707: First names of the authors should be replaced by initials.

c) Aas, W., Eckhardt, S., Fiebig, M., Solberg, S., and Yttri, K. E.: Monitoring of long-range transported air pollutants in Norway, annual report 2019. Kjeller, NILU (Miljødirektoratet rapport, M-1710/2020) (NILU OR, 4/2020), 2020.

Lines 959-968: "Myhre and Samset, 2015" should come before "Myhre et al., 2013".

d) Myhre and Samset (2015) placed above Myhre et al. (2013) in reference list.

Lines 1067-1070 should be deleted.

e) Lines 1067-1070 has been deleted.

The following reference of the text are missing in the Reference list: Line 67: Pio et al., 2007; f) The following reference has been included: Pio, C. A., Legrand, M., Oliveira, T., Afonso, J., Santos, C., Caseiro, A., Fialho, P., Barata, F., Puxbaum, H., Sanchez-Ochoa, A., Kasper-Giebl, A., Gelencser, A., Preunkert, S., and Schock, M.: Climatology of aerosol composition (organic versus inorganic) at nonurban sites on a west-east transect across Europe, J. Geophys. Res.-Atmos., 112, 10.1029/2006jd008038, 2007.

Line 108: Sillanpää et al., 2006; there is "Sillanpää et al., 2005" in the Reference list, but there is not referred to this within the text. g) The correct reference is Sillanpaa et al. (2005) thus Sillanpaa et al. (2005) is included in the reference list: Sillanpaa, M., Frey, A., Hillamo, R., Pennanen, A. S., and Salonen, R. O.: Organic, elemental and inorganic carbon in particulate matter of six urban environments in Europe, Atmos. Chem. Phys., 5, 2869-2879, 10.5194/acp-5-2869-2005, 2005. and Sillanpää et al. (2006) has been changed to Sillanpää et al. (2005) in the text.

The following reference of the text are missing in the Reference list: Line 109: Genberg et al., 2013. h) Genberg et al., 2013 should be Genberg et al., 2011, which is included in the reference list. Genberg et al. (2013) has been changed to Genberg et al. (2011) in the text.

The following reference of the text are missing in the Reference list: Line 176: Mann, 1945 i) The following reference has been included: Mann, H. B.: Non-parametric tests against trend, Econometrica 13:163-171, 1945.

The following reference of the text are missing in the Reference list: Lines 177 and 178: Gilbert, 1987 j) The following reference has been included: Gilbert, R. O.: Statistical Methods for Environmental Pollution Monitoring, Wiley, NY, United States, pp. 336, 1987.

The following reference of the text are missing in the Reference list: Line 178: Theil, 1958; perhaps this should be "1950" instead of "1958, see below.

k) Should be Theil, 1950, which is correct in the reference list. Theil, H.: A rank-invariant method of linear and polynomial regression analysis. Proc. R. Netherlands, Acad. Sci. 53, 386–392, https://doi.org/10.1007/978-94-011-2546-8_20, 1950. and Theil et al. (1958) has been changed to Theil et al. (1950) in the text.

The following reference of the text are missing in the Reference list: Line 178: Sen. 1968. l) The following reference has been included: Sen, P. K.: Estimates of the regression coefficient based on Kendall's Tau. J. Am., Stat. Assoc. 63 (324), 1379–1389.https://doi.org/10.2307/2285891, 1968.

The following reference of the text are missing in the Reference list: Line 215: Polissar et al., 1998. m) The following reference has been included: Polissar, A. V., Hopke, P. K., and Paatero, P.: Atmospheric aerosol over Alaska - 2. Elemental composition and sources, J. Geophys. Res.-Atmos., 103, 19045-19057, 10.1029/98jd01212, 1998.

The following reference of the text are missing in the Reference list: Line 215: Norris et al., 2014. n) The following reference has been included: Norris, G., Duvall, R., Brown, S., and Bai, S.: EPA Positive Matrix Factorization (PMF) 5.0 Fundamentals and User Guide, U.S. Environmental Protection Agency, Washington, DC, 20460 (i-124, EPA/600/R-14/108, April), 2014.

The following reference of the text are missing in the Reference list: Line 241: Pacyna et al., 1996.  o) Pacyna et al.  (1996) should be Pacyna et al.  (1986).  Pacyna et al.  (1986) is present in the reference list.  We have changed Pacyna et al.  (1996) to Pacyna et al.  (1986) in the text.

Line 305: Stumm and Mrogan, 1995; there is "Stumm and Morgan, 1996" in the Reference list, but there is not referred to this within the text.  p) Should be Stumm and Morgan, 1996, which is present in the reference list.  Stumm and Morgan (1995) has been changed to Stumm and Morgan (1996) in the text.

The following reference of the text are missing in the Reference list:  Line 620: Spracklen et al., 2011. q) The following reference has been included: Spracklen, D. V., Jimenez, J. L., Carslaw, K. S., Worsnop, D. R., Evans, M. J., Mann, G. W., Zhang, Q., Canagaratna, M. R., Allan, J., Coe, H., McFiggans, G., Rap, A., and Forster, P.: Aerosol mass spectrometer constraint on the global secondary organic aerosol budget, Atmos. Chem. Phys., 11, 12109-12136, 10.5194/acp-11-12109-2011, 2011.

There is no reference made in the Main text to the following references that are in the Reference list: Aas et al., 2019. r) Aas et al., 2019 should be in Suppl. only and have been removed from the reference list of the main text.

There is no reference made in the Main text to the following references that are in the Reference list:  Birch and Cary, 1996 s) Birch and Cary, 1996 do not appear in the paper and has been removed from the reference list.

There is no reference made in the Main text to the following references that are in the Reference list: Cattiaux et al., 2010. t) Cattiaux et al., 2010 do not appear in the paper and has been removed from the reference list.

There is no reference made in the Main text to the following references that are in the Reference list: Fine et al, 2001; 2002a, 2002b, 2004. u) Fine et al, 2001; 2002a, 2002b, 2004 do not appear in the paper and has been removed from the reference list.

There is no reference made in the Main text to the following references that are in the Reference list: Hu et al, 2018. v) Hu et al., 2018 do not appear in the paper and has been removed from the reference list.

There is no reference made in the Main text to the following references that are in the Reference list: Jordan and Seen, 2005. w) Jordan and Seen, 2005 do not appear in the paper and has been removed from the reference list.

There is no reference made in the Main text to the following references that are in the Reference list: Long et al., 2013. x) Long et al., 2013 do not appear in the paper and has been removed from the reference list.

There is no reference made in the Main text to the following references that are in the Reference list: Putaud et al., 2010. y) Putaud et al., 2010 do not appear in the paper and has been removed from the reference list.

There is no reference made in the Main text to the following references that are in the Reference list: Schmidl et al., 2008. z) Schmidl et al., 2008 do not appear in the paper and has been removed from the reference list.

There is no reference made in the Main text to the following references that are in the Reference list: Theobald et al., 2019. aa) Theobald et al., 2019 should be in Suppl. only and have been removed from the reference list of the main text.

There is no reference made in the Main text to the following references that are in the Reference list: Turpin and Lim., 2001. bb) Turpin and Lim, 2001 do not appear in the paper and has been removed from the reference list.

There is no reference made in the Main text to the following references that are in the Reference list: Torseth et al., 2000. cc) Torseth et al., 2000 do not appear in the paper and has been removed from the reference list.

Winiwarter et al., 1999 do not appear in the paper and has been removed from the reference list. dd) Winiwarter et al., 1999 do not appear in the paper and has been

removed from the reference list.

References that are mentioned in its text should also be included in the Reference list, even if they are already in the Reference list of the Main text.

Line 230: The title of the journal article should be included.

ee) The title has been included: Alier, M., van Drooge, B. L., Dall'Osto, M., Querol, X., Grimalt, J. O., and Tauler, R.: Source apportionment of submicron organic aerosol at an urban background and a road site in Barcelona (Spain) during SAPUSS, Atmos. Chem. Phys., 13, 10353-10371, 10.5194/acp-13-10353-2013, 2013.

Lines 233 and 254: Is it Backman or Backmann? In lines 116 and 118 there is Backmann, but in line 154 Backman

ff) It should be one "n" and this has been corrected: Backman, J., Schmeisser, L., Virkkula, A., Ogren, J. A., Asmi, E., Starkweather, S., Sharma, S., Eleftheriadis, K., Uttal, T., Jefferson, A., Bergin, M., Makshtas, A., Tunved, P., and Fiebig, M.: On Aethalometer measurement uncertainties and an instrument correction factor for the Arctic, Atmos. Meas. Techn., 10, 5039-5062, 10.5194/amt-10-5039-2017, 2017.

The following references of the text are missing in the Reference list: Line 7: Yttri et al., 2007a;

gg) The following reference has been included: Yttri, K. E., Aas, W., Bjerke, A., Cape, J. N., Cavalli, F., Ceburnis, D., Dye, C., Emblico, L., Facchini, M. C., Forster, C., Hanssen, J. E., Hansson, H. C., Jennings, S. G., Maenhaut, W., Putaud, J. P., and Torseth, K.: Elemental and organic carbon in PM10: a one year measurement campaign within the European Monitoring and Evaluation Programme EMEP, Atmos. Chem. Phys., 7, 5711-5725, 2007a.

The following references of the text are missing in the Reference list: Line 10: Yttri et al., 2011; also, this should either be 2011a or 2011b or even 2011c;
hh) The following reference has been included in the Suppl. reference list and marked "Yttri et al., 2011b" in the Suppl. text: Yttri, K. E., Simpson, D., Nojgaard, J. K., Kristensen, K., Genberg, J., Stenstrom, K., Swietlicki, E., Hillamo, R., Aurela, M., Bauer, H., Offenberg, J. H., Jaoui, M., Dye, C., Eckhardt, S., Burkhart, J. F., Stohl, A., and Glasius, M.: Source apportionment of the summer time carbonaceous aerosol at Nordic rural background sites, Atmos. Chem. Phys., 11, 13339-13357, 10.5194/acp-11-13339-2011, 2011b.

The following references of the text are missing in the Reference list: Lines 10-11: Yttri et al., 2019;

ii) The following reference has been included in the Suppl. Reference list: Yttri, K. E., Simpson, D., Bergstrom, R., Kiss, G., Szidat, S., Ceburnis, D., Eckhardt, S., Hueglin, C., Nojgaard, J. K., Perrino, C., Pisso, I., Prevot, A. S. H., Putaud, J. P., Spindler, G., Vana, M., Zhang, Y. L., and Aas, W.: The EMEP Intensive Measurement Period campaign, 2008-2009: characterizing carbonaceous aerosol at nine rural sites in Europe, Atmos. Chem. Phys., 19, 4211-4233, 10.5194/acp-19-4211-2019, 2019.

The following references of the text are missing in the Reference list: Line 18: Cavalli et al., 2010;

jj) The following reference has been included in the Suppl. Reference list: Cavalli, F., Viana, M., Yttri, K. E., Genberg, J., and Putaud, J.-P.: Toward a standardised thermal-optical protocol for measuring atmospheric organic and elemental carbon: the EUSAAR protocol, Atmos. Meas. Tech., 3, 79-89, 2010.

Line 23: Subramanian et al., 2006; there is "Subramanian et al., 2004" in the Reference list, but there is not referred to this within the text

kk) Should be Subramanian et al., 2004 in the text. This has been corrected. Reference in Suppl. ref. list is correct.

Line 57: Theil, 1958; There is "Theil, 1950" in the Reference list, and it looks that this

is the correct year.

ll) Should be Theil, 1950. This has been corrected. Reference in Suppl. ref. list is correct.

The following references of the text are missing in the Reference list: Tørseth et al., 2012.

mm) The following reference has been included in the Suppl. Reference list: Torseth, K., Aas, W., Breivik, K., Fjaeraa, A. M., Fiebig, M., Hjellbrekke, A. G., Myhre, C. L., Solberg, S., and Yttri, K. E.: Introduction to the European Monitoring and Evaluation Programme (EMEP) and observed atmospheric composition change during 1972-2009, Atmos. Chem. Phys., 12, 5447-5481, 10.5194/acp-12-5447-2012, 2012.

The following references of the text are missing in the Reference list: Line 61: Aas et al. 2019. nn) The following reference has been included in the Suppl. Reference list: Aas, W., Mortier, A., Bowersox, V., Cherian, R., Faluvegi, G., Fagerli, H., Hand, J., Klimont, Z., Galy-Lacaux, C., Lehmann, C. M. B., Myhre, C. L., Myhre, G., Olivie, D., Sato, K., Quaas, J., Rao, P. S. P., Schulz, M., Shindell, D., Skeie, R. B., Stein, A., Takemura, T., Tsyro, S., Vet, R., and Xu, X. B.: Global and regional trends of atmospheric sulfur (vol 9, 953, 2019), Sci. Rep., 10, 10.1038/s41598-020-62441-w, 2020.

The following references of the text are missing in the Reference list: Line 125: Sandradewi et al. 2008. oo) The following reference has been included in the Suppl. Reference list: Sandradewi, J., Prevot, A. S. H., Szidat, S., Perron, N., Alfarra, M. R., Lanz, V. A., Weingartner, E., and Baltensperger, U.: Using aerosol light absorption measurements for the quantitative determination of wood burning and traffic emission contributions to particulate matter, Environ. Sci. Technol., 42, 3316-3323, 10.1021/es702253m, 2008.

Line 137: Paatero et al., 1994; there is "Paatero and Tapper, 1994" in the Referencelist, but there is not referred to this within the text.

pp) Paatero et al., 1994 has been changed to "Paatero and Tapper, 1994" in the text. Thus, the reference in the Suppl. ref. list is correct.

The following references of the text are missing in the Reference list: Zotter et al., 2014

qq) The following reference has been included in the Suppl. Ref. list: Zotter, P., Ciobanu, V. G., Zhang, Y. L., El-Haddad, I., Macchia, M., Daellenbach, K. R., Salazar, G. A., Huang, R. J., Wacker, L., Hueglin, C., Piazzalunga, A., Fermo, P., Schwikowski, M., Baltensperger, U., Szidat, S., and Prevot, A. S. H.: Radiocarbon analysis of elemental and organic carbon in Switzerland during winter-smog episodes from 2008 to 2012-Part 1: Source apportionment and spatial variability, Atmos. Chem. Phys., 14, 13551-13570, 10.5194/acp-14-13551-2014, 2014.

Line 221: Bauer et al., 2008a (incidentally, the "a" should be removed).

rr) The "a" in Bauer et al., 2008a has been removed.

Line 221: Yttri et al., 2007a;)

ss) Yttri et al 2007a has been changed to Yttri et al., 2007b. The reference can be found in the Suppl. ref. list

Lines 221-222: Yttri et al., 2011a,b.

tt) The Yttri et al. 2011a and Yttri et al., 2011b references have been added to the Suppl. ref. list.:

Yttri, K. E., Simpson, D., Nojgaard, J. K., Kristensen, K., Genberg, J., Stenstrom, K., Swietlicki, E., Hillamo, R., Aurela, M., Bauer, H., Offenberg, J. H., Jaoui, M., Dye, C., Eckhardt, S., Burkhart, J. F., Stohl, A., and Glasius, M.: Source apportionment of the summer time carbonaceous aerosol at Nordic rural background sites, Atmos. Chem. Phys., 11, 13339-13357, 10.5194/acp-11-13339-2011, 2011b.

Yttri, K. E., Simpson, D., Stenstrom, K., Puxbaum, H., and Svendby, T.: Source apportionment of the carbonaceous aerosol in Norway - quantitative estimates based on

C-14, thermal-optical and organic tracer analysis, Atmos. Chem. Phys., 11, 9375-9394, 10.5194/acp-11-9375-2011, 2011a.

The Pio et al., 2007 reference appear in the reference list but not in the text.

uu) The Pio et al., 2007 reference has been removed from the Suppl. ref. list. Minor comments by #R1: Minor comments and corrections for the Main Text: Line 21: Replace "organic-" by "organic". a) "Organic-" was replaced by "organic" Line 23: Replace "-at the" by "at the". b) "-at the" was replaced by "at the". Lines 27-30: It says "six" in line 27, but then in the remainder of the sentence 7 components are listed. This is confusing. c) We have rephrased the original sentence to so that it does not cause confusion to the reader: New version: "Using positive matrix factorization (PMF) we identify seven carbonaceous aerosol sources at Birkenes: Mineral dust dominated (MIN), traffic/industry-like (TRA/IND), short range transported biogenic secondary organic aerosol (BSOASRT), primary biological aerosol particles (PBAP), biomass burning (BB), ammonium nitrate dominated (NH4NO3), and (one low carbon fraction) sea salt (SS)." Original version: "Using positive matrix factorization (PMF) we identify six carbonaceous aerosol sources at Birkenes: Mineral dust dominated (MIN), traffic/industry-like (TRA/IND), short range transported biogenic secondary organic aerosol (BSOASRT), primary biological aerosol particles (PBAP), biomass burning (BB), and ammonium nitrate dominated (NH4NO3), and one low carbon fraction sea salt (SS)." Line 72: Replace "and fungus" by "and fungi". d) "fungus" was replaced by "and fungi". Line 90: Replace "biogenic," by "biogenic;". e) "biogenic," was replaced by "biogenic;". Line 118: Replace "have developed" by "has developed". f) "have developed" was replaced by "has developed". Line 170: Replace "monosaccharides anhydrides" by "monosaccharide anhydrides". g) "monosaccharides anhydrides" was replaced by "monosaccharide anhydrides". Line 182: Replace "Scientific" by "Scientific)". h) "Scientific" was replaced by "Scientific)". Line 184: Replace "in which" by "which". i) "in which" was replaced by "which". Line 209: Replace "principal" by "principle". j) "principal" was replaced by "principle". Line 217: Replace "is via" by

"are via". k) "is via" was replaced by " Source apportionment by PMF is ". Line 292: Replace "also favours" by "also favour". l) "also favours" was replced by "also favour". Line 305: Replace "resembles these" by "resemble these". m) "resembles these" was replaced by "resemble these". Line 334: Replace "EC, (Sect. 3.1.2) but" by "EC (Sect. 3.1.2), but". n) "EC, (Sect. 3.1.2) but" was replaced by "EC (Sect. 3.1.2), but". Line 349: Replace "Table S 8" by "Table S 8)". o) "Table S 8" was replaced by "Table S 8)". Line 363: The reference to Table S 4 here is unclear; I presume that reference should be made to another Table. p) "Table S 4" was changed to "Table S 5". Lines 395-396: The data listed here apparently relate to PM2.5 and PM10; it should be specified which ones are for which size fraction. Also in line 396, replace "-1.8, CI"by "-1.8%, CI". q) New version: Furthermore, and although one should be careful drawing conclusions from non-significant outcomes, it is worth noting that the levoglucosan to EC ratio most likely increased (+2.8% yr-1( PM10), CI = -3.5 − +6.5% yr-1 and +2.3% yr-1 (PM2.5), CI = -2.2 − 5.0 % yr-1) for the period 2008–2018, whereas it most likely decreased (-1.8% yr-1 (PM10), CI = -10.6 − +1.8 and -3.6% yr-1 (PM2.5), CI = -9.8 − +1.3% yr-1) for the levoglucosan to OC ratio (Table S 13). Original version: Furthermore, and although one should be careful drawing conclusions from non-significant outcomes, it is worth noting that the levoglucosan to EC ratio most likely increased (+2.8% yr-1, CI = -3.5 − +6.5% yr-1 and +2.3% yr-1, CI = -2.2 − 5.0 % yr-1) for the period 2008–2018, whereas it most likely decreased (-1.8%, CI = -10.6 − +1.8 and -3.6% yr-1, CI = -9.8 − +1.3% yr-1) for the levoglucosan to OC ratio (Table S 13). Line 455: Replace "et al.," by "et al.". r) "et al.," was replaced by "et al.". Line 472: Replace "airmasses" by "air masses". s) "airmasses" was replaced by "air masses". Line 530: Replace "2003)" by "2003". t) "2003)" was replaced by "2003". Line 605: Replace "points to" by "point to". u) "points to" was replaced by "point to". Line 619: Replace "(2012) but" by "(2012), but". v) "(2012) but" was replaced by "(2012), but". Line 634: Replace "likely differ" by "likely differs". w) "likely differ" was replaced by "likely differs". Line 698: Replace "I.e." by "i.e.". x) "I.e." was replaced by "i.e.". Line 714: Replace "Sci Rep" by "Sci. Rep.". y) The Aas et al. (2019) ref., in which Sci

Rep occurs should be in the Suppl ref, list only, not in the main text ref. list. Line 786: Replace "organicaerosols" by "organic aerosols". z) "organicaerosols" was replaced by "organic aerosols". Line 904: Replace "5," by "5, 5065,". aa) According to EndNote and the Copernicus format, Hodnebrog et al. (2014) should be: Hodnebrog, O., Myhre, G., and Samset, B. H.: How shorter black carbon lifetime alters its climate effect, Nat. Commun., 5, 10.1038/ncomms6065, 2014. as stated in the reference list of the revised manuscript. Line 970: Replace "O'dowd" by "O'Dowd". bb) "O'dowd" was replaced by "O'Dowd". Line 1019: Replace "Sioutas" by "and Sioutas". cc) "Sioutas" was replaced by "and Sioutas". Line 1106: Replace "B.J., Lim" by "B. J. and Lim" in case this reference is retained. dd) The reference was not retained. Line 1120: Replace "www.atmos-chem-phys.net/12/5447/2012/doi:10.5194/acp-12-5447-2012," by https://doi.org/10.5194/ acp-12-5447-2012, ee) We made the change to the reference as pointed out by R1:

Torseth, K., Aas, W., Breivik, K., Fjaeraa, A. M., Fiebig, M., Hjellbrekke, A. G., Myhre, C. L., Solberg, S., and Yttri, K. E.: Introduction to the European Monitoring and Evaluation Programme (EMEP) and observed atmospheric composition change during 1972-2009, Atmos. Chem. Phys., 12, 5447-5481, 10.5194/acp-12-5447-2012, 2012. Line 1139: Replace "pri-mary" by "primary". ff) "pri-mary" was replaced by "primary". Line 1187: Replace "Atmos. Chem. Phys. Discuss., 14, pp. 15591-15643," by "Atmos. Chem. Phys., 14, 13551-13570, https://doi.org/10.5194/acp-14-13551-2014,". gg) We made the change to the reference as pointed out by R1: Zotter, P., Ciobanu, V. G., Zhang, Y. L., El-Haddad, I., Macchia, M., Daellenbach, K. R., Salazar, G. A., Huang, R. J., Wacker, L., Hueglin, C., Piazzalunga, A., Fermo, P., Schwikowski, M., Baltensperger, U., Szidat, S., and Prevot, A. S. H.: Radiocarbon analysis of elemental and organic carbon in Switzerland during winter-smog episodes from 2008 to 2012-Part 1: Source apportionment and spatial variability, Atmos. Chem. Phys., 14, 13551-13570, 10.5194/acp-14-13551-2014, 2014. Lines 1192-1193: Replace "www.atmos-chem-phys.net/17/4229/2017/doi:10.5194/acp-17-4229-2017," by "https://doi.org/10.5194/ acp-17-4229-2017,". hh) We made the change to

the reference as pointed out by R1: Zotter, P., Herich, H., Gysel, M., El-Haddad, I., Zhang, Y. L., Mocnik, G., Huglin, C., Baltensperger, U., Szidat, S., and Prevot, A. H.: Evaluation of the absorption angstrom ngstrom exponents for traffic and wood burning in the Aethalometer-based source apportionment using radiocarbon measurements of ambient aerosol, Atmos. Chem. Phys., 17, 4229-4249, 10.5194/acp-17-4229-2017, 2017. Line 1215: Replace "); CA" by "; CA". ii) "); CA" was replaced by "; CA". Line 1251, within Table 2: Replace twice "Zotter et al. 2017" by "Zotter et al., 2017" ii) "Zotter et al. 2017" was replaced by "Zotter et al., 2014"

Minor corrections for the Supplement: Line 41: Replace "is text not" by "text are not". a) "is text not" was replaced by "text are not". Line 64: Replace "often deviates" by "often deviate". b) "often deviates" was replaced by "often deviate". Line 83: Replace "was applied" by "were applied". c) "was applied" was replaced by "were applied". Line 116: Replace "by of" by "by". d) "by of" was replaced by "by". Line 117: Replace "adapt the" by "adopt the". e) "adapt the" was replaced by "adopt the". Line 194: Replace "then then" by "then". f) "then then" was replaced by "then". Line 214: Replace ") , and" by "), and". g) ") , and" was replaced by "), and". (We found this in line 195, not 214) Line 240: Replace "2013" by "2013.". h) "2013" was replaced by "2013.". Line 343: Replace "1389.https" by "1389, https". i) "1389.https" was replaced by "1389, https". Line 394: Replace "bar (Figure S 2)" by "bar". j) "bar (Figure S 2)" was replaced by "bar". Line 396: Replace "were only" by "was only". k) "were only" was replaced by "was only". Line 404: Replace "for their identification" by "for identification". l) "for their identification" was replaced by "for identification". Line 404, within Table S 1: Replace "Methylterythritol" by "Methylerythritol". m) "Methylterythritol" was replaced by "Methylerythritol".

Action: We have made the following changes and additions to the paper separate from the comments made by Referee #1 and #2 and the short comment made by Martin Schulz:

We realized that the title of our revised manuscript could be improved, making the

following change: "Trends, composition, and sources of carbonaceous aerosol at the Birkenes Observatory, Northern Europe, 2001-2018. Hence, it would be possible for a future reader to know the time span the paper covers by just reading the title. For sure, the phrase "last 18 years", used in the original title is imprecise.

"TRA/IND" was stated the other way around (IND/TRA) three places in the manuscript, we have changed this so that all reads "TRA/IND".

We have switched from "Positive Matrix Factorisation" to "Positive Matrix Factorization" due to the US origin of this analytical approach and for consistency throughout the manuscript.

---

## Author Comment (AC3) · 18 Mar 2021

Reply to referee #2: The present study investigates the carbonaceous composition of PM2.5 and PM10 obtained at the Birkenes Observatory (GAW - EMEP) from 2001 to 2018. The data series is unique in Europe and has invaluable scientific interest. The treatment and interpretation of the data is adequate, and the results and conclusions obtained are very relevant. The results demonstrated a long-term change in the chemical composition of the aerosol at the background site of Birkenes and therefore also in the sources contribution to PM. Authors applied PMF receptor model to the 2016-2018 chemical dataset identifying 6 carbonaceous aerosol sources. They

demonstrated a decrease of traffic and industry OC/EC emissions, while the abatement of OC/EC from biomass burning has been slightly less successful. Moreover, results emphasize the importance of biogenic sources (BSOA and PBAP) at Birkenes site. The results demonstrated a decrease in OC/EC emissions from traffic and industry affecting the Birkenes site. This decrease is not as obvious for OC / EC from biomass burning, indicating a less successful abatement strategy. The results also emphasize the importance of biogenic sources (BSOA and PBAP) at the Birkenes site. Authors concluded the need of investigating trends in levoglucosan and biomass burning in Europe, given the further importance residential wood burning as a major source of air pollution in Europe. The need of measuring monoterpene and sesquiterpene oxidation products for improving the SOA apportionment is also highlighted. These data can be essential to improve the model outputs.

Reply to minor correction by referee #2:

We would like to thank referee 2 (R2) for his/her work valuable comments and corrections to our manuscript.

The data regarding the decreasing EC fractions in PM2.5 (-4.0% yr-1) and PM10 (-4.7% yr-1) in the abstract differs from the data in the main text (Line380: whereas it decreased for the EC fraction (-3.9 – -4.5% yr-1).

a) We have changed the values for the EC fraction of PM10 and PM2.5 in the abstract to "PM2.5 (-3.9% yr-1)" and PM10 (-4.5% yr-1) in line with the notification made by R2.

Line 58. I think OC and EC are not regularly measured in Air Quality Monitoring networks, with the exception of EMEP/GAW and aerosol in-situ ACTRIS sites.

b) We have changed the sentence to the following ". . .. . . is measured regularly in major air monitoring networks such as e.g., EMEP and IMPROVE (e.g., Malm et al., 1994; Tørseth and Hov, 2003; Tørseth et al., 2012)." The Malm et al. 1994 reference has been added to the reference list.

Method section 2.2.1. What is the difference between the old and the new Birkenes Observatory? When was the new observatory installed? Please add some comments on 2.1.2,2,2 –

c) We have added the following text to include that there was a shift from the old Birkenes station to the new Birkenes Observatory in 2009:

"The Birkenes Observatory (58°23'N, 8°15'E, 219 m above sea level, asl) is an EMEP/GAW (Global Atmospheric Watch) supersite in southern Norway (Figure 1) situated 100 m south-east of the old Birkenes site, initiating measurements in 2009."

Lines 187 188- It should be noted here that the "novel" positive matrix factorization for BC data is based on the methodology devised by Platt et al, in preparation. Is there any other reference to this method available (e.g., conference proceedings....)?

d) There is not yet any (other) reference available for Platt et al. (in prep.). Platt et al. (in prep.) has been added to the text.

2.4. It should be noted here that the PMF was applied to the 2016-2018 dataset.

e) We have modified the first sentence of Chapter 2.4 to account for the comment made by R2.

"We performed PMF ME2 (Canonaco et al., 2013) (See Sect. S3 for a description of the analysis principal and S4 for its application to filter data) for samples collected in 2016-2018 (151 samples),.."

Line210: was OC in PM2.5-10 calculated by difference (OCPM10 – OC PM2.5)? Was EC in PM25 equal (or similar) to EC in PM10?

f) Yes, OC in PM10-2.5 was calculated by the difference between OC in PM10 and OC in PM2.5.

Concerning, whether EC in PM2.5 was equal (or similar) to EC in PM10: In Line 321-322 we write that: "EC, being from combustion that generates fine PM, was almost

exclusively associated with PM2.5, ….". In line 565-566 we write: "In the present study, low levels of coarse fraction EC occasionally appear in summer and fall (Table S5), following the seasonality of PBAP". In line 566-559, we argue that coarse EC most likely is an analytical artifact, resulting from charring of some types of PBAP dusting thermal-optical analysis.

---

## Author Comment (AC4) · 18 Mar 2021

Comment by Michael Schulz:

The work by Yttri et al. presents a great, valuable summary of 18 years of organic aerosol measurements. The absent organic carbon (OC) trend is most interesting to me. I believe this could be highlighted even more in abstract and conclusions. What do other OC trends show in Europe? Is this the only work attempting to calculate OC trends at European sites? Could the present work be put a bit more in a European context? More references? A side question: What influence has the change of the OC

measurement protocol in the middle of the period on the OC trend? In what direction can this have influenced the trend?

We would like to thank Dr Schulz for his interest in our manuscript!

Dr Schulz states that: "The absent organic carbon (OC) trend is most interesting to me. I believe this could be highlighted even more in abstract and conclusions".

a) We agree that the lack of a statistically significant downward trend in the OC concentration is an interesting finding. One would expect a reduction of anthropogenic OC that reflects that of EC, at least for primary OC, but probably also SOA from anthropogenic precursors. In our manuscript, we argue that this reduction is not visible, as OC is dominated by emissions from natural sources (Biogenic SOA and PBAP). Note though that OCPM2.5, which has as larger fraction of anthropogenic OC compared to PM10-2.5 and PM10, has a minor downward "trend" (-0.8% yr-1), although not statistical significant.

In the abstract we have included the following sentence to meet the request from Michael Schulz.

"Dominating biogenic sources explain why there was no downward trend for OC."

In our conclusions, line 652 – 657 focus on biogenic sources as an explanation to why no decreasing trend was observed for OC. We have added the following sentence to underpin the importance of such long time series for carbonaceous aerosol:

"We emphasize the importance of establishing long lasting, high quality carbonaceous aerosol and organic tracers time series at several sites across Europe for this purpose."

• What do other OC trends show in Europe? Is this the only work attempting to calculate OC trends at European sites? Could the present work be put a bit more in a European context? More references?

b) To our knowledge there are no other studies performing trend analysis of OC in

Europe based on measurements; the present study is the first one. We expect that within a few years, time series of equal length as that of Birkens at present will be available for 1-2 southern European sites. Consequently, there are no other studies to compare with, and hence not possible to put our results into a European context by comparing with other time-series. Towards the end of the introduction (line 124 - 125), we emphasize that the Birkenes Observatory is well suited to monitor air pollution from Continental Europe, and in the third line of the abstract (line 23-24) we state that Birkenes is a site representative of the Northern European region, thus the connection to the greater Europe is established.

What influence has the change of the OC measurement protocol in the middle of the period on the OC trend? In what direction can this have influenced the trend?

c) In the Suppl., line 35-42 we write the following:

A comparison of the two temperature programmes (denoted "protocol" by Michael Schulz) used for the Birkenes time series was performed for PM2.5 filter samples collected at Birkenes in 2014, using temperature calibrated versions of both Quartz and EUSAAR-2. There was a good agreement between the two temperature programs for TC and OC, i.e., close to the expected uncertainty associated with analysis and sampling, whereas for EC the difference was pronounced (Table S 17), although in close correspondence with that previously reported by Panteliadis et al. (2015). Note that OC and EC data for the period 2001–2007 discussed in the main is text not corrected according to Eq. (S 18–20) (Table S 17), except for the purpose of trend calculations.

Hence, the (minor) difference in OC obtained by the Quartz and EUSAAR-2 temperature programmes (See Table S 17, Eq. S 18-20) is accounted for in the trend calculations that are performed.